# Diet changes due to urbanization in South Africa are linked to microbiome and metabolome signatures of Westernization and colorectal cancer

Transition from traditional high-fiber to Western diets in urbanizing communities of Sub-Saharan Africa is associated with increased risk of non-communicable diseases (NCD), exemplified by colorectal cancer (CRC) risk. To investigate how urbanization gives rise to microbial patterns that may be amenable by dietary intervention, we analyzed diet intake, fecal 16 S bacteriome, virome, and metabolome in a cross-sectional study in healthy rural and urban Xhosa people (South Africa). Urban Xhosa individuals had higher intakes of energy (urban: 3,578 ± 455; rural: 2,185 ± 179 kcal/d), fat and animal protein. This was associated with lower fecal bacteriome diversity and a shift from genera favoring degradation of complex carbohydrates (e.g., *Prevotella*) to taxa previously shown to be associated with bile acid metabolism and CRC. Urban Xhosa individuals had higher fecal levels of deoxycholic acid, shown to be associated with higher CRC risk, but similar short-chain fatty acid concentrations compared with rural individuals. Fecal virome composition was associated with distinct gut bacterial communities across urbanization, characterized by different dominant host bacteria (urban: Bacteriodota; rural: unassigned taxa) and variable correlation with fecal metabolites and dietary nutrients. Food and skin microbiota samples showed compositional differences along the urbanization gradient. Rural-urban dietary transition in South Africa is linked to major changes in the gut microbiome and metabolome. Further studies are needed to prove cause and identify whether restoration of specific components of the traditional diet will arrest the accelerating rise in NCDs in Sub-Saharan Africa.

Worldwide, colorectal cancer (CRC) is the second most common cause of cancer in women and the third in men[1]. There is a remarkable geographical variation in CRC incidence, with the highest rates in North America, Australia/New Zealand, and Europe (all >40:100,000) and the lowest in rural Africa (<5:100,000). Convincing evidence supports the now classic observations of Burkitt, working in East Africa, and Walker in South Africa in the 60s–80s, which demonstrated an inverse association between CRC risk and traditional African diets rich in fiber[2]. Recent large-scale prospective studies and meta-analyses confirmed that a high intake of dietary fiber correlates with low CRC risk[3,4], whereas consumption of ultra-processed foods low in fiber is linked to higher CRC rates[5]. Dietary fiber is fermented by gut bacteria

e-mail: sjokeefe@pitt.edu

to short-chain fatty acids (SCFA), including butyrate, which serves as the chief source of energy for the colonic epithelium and has anti-inflammatory and anti-neoplastic effects[6,7]. Fiber also reduces intestinal exposure to dietary carcinogens by lowering the colonic transit time and binding tumor-promoting secondary bile acids[8], collectively promoting gut microbiota homeostasis. In addition to the critical importance of dietary fiber in maintaining colonic health, recent studies highlighted the implication of a high-fiber diet to reduce disease risk beyond CRC: the intake of fiber is associated with decreased all-cause mortality and lower risk of non-communicable diseases (NCD) such as diabetes, cardiovascular diseases and obesity[4,9]. Being an upper-middle-income country, South Africa faces high rates of NCDs that pose a massive threat with increasing urbanization and started to extend even to the rural segments of the population[10–12]. About 63% of South Africans are already living in urban areas, and the number is expected to rise to 71% by 2030[13]. As part of this ongoing urbanization process, the adoption of Western dietary patterns will promote the rapid increase of NCDs throughout Sub-Saharan Africa. Thus, NCDs are set to become the leading cause of disability and mortality in this region and must be recognized as major barriers to attaining the African Sustainable Development Goals[14,15]. Here, we investigate how the ongoing transition from traditional to urban lifestyle affects the gut bacteriome, virome and its metabolome in samples of the second largest section of the black population (amaXhosa) in South Africa to identify signatures that might be amenable to change through dietary intervention.

## Results

### Urban individuals have a Western dietary pattern associated with lower diversity and distinct composition of the gut microbiota

We performed a cross-sectional, observational pilot study in two healthy middle-aged rural or urban South African amaXhosa (Xhosa) cohorts. The urban cohort ($n = 20$) was located in Cape Town, Western Cape Province, while the rural group ($n = 24$) was recruited in the rural Eastern Cape Province of South Africa (Table 1). Food consumption was higher in the urban Xhosa cohort, with a mean energy intake of 3,578 kcals/d in urban compared to 2185 kcal/d in rural study participants (Table 2). Analysis of the contribution of carbohydrate, fat, and protein to total energy indicated that the composition was within normal limits corresponding to Western diets for the urban Xhosa cohort, whereas rural individuals consumed a high carbohydrate diet providing 72% of calories, whilst fat only contributed 16.9% of total calories. This translated into a higher absolute intake of dietary fiber in urban individuals but normalized to 1000 kcals, the rural Xhosa cohort had a slightly higher fiber intake (Table 2). Consistent with their less Westernized dietary pattern, the rural Xhosa diet was characterized by a lower intake of cholesterol and animal protein, as well as higher amounts of plant protein when normalized to kcals/d intake (Table 2). Similarly, dietary diversity and polyphenol consumption were higher in the rural Xhosa cohort because of a higher amount, variety, and frequency of plant foods eaten (e.g., 'imifino', onion, cabbage, and pumpkin leaves).

For fecal microbiota analysis by 16S rRNA gene sequencing, we noted lower α-diversity indices in urban samples (Fig. 1A) and a distinct compositional clustering based on generalized UniFrac distances according to rural–urban status (Fig. 1B). Despite no clear separation of the two cohorts at phylum level, regional sub-clusters can be identified by high abundance of Bacteriodota (rural) or Proteobacteria (urban) (Fig. 1C). Urban fecal samples were characterized by high numbers of operational taxonomic units (OTU) related to *Bacteroides*, and other genera such as *Lachnoclostridium*, *Fusobacterium*, *Alistipes*, and *Bilophila* (Fig. 1D). Despite the similar fiber intake, a higher abundance of OTUs assigned to *Prevotella*, *Faecalibacterium*, *Dialister* that are involved in saccharolytic fermentation and SCFA production was

**Table 1 | Demographic data of study participants**

| Variable | Rural | Urban | *p* Value |
|---|---|---|---|
| Geographic area | Zithulele, Eastern Cape | Khayelitsha, Cape Town | |
| Number of participants (n) | 24 | 20 | |
| Age (mean ± SD) | 56.7 ± 11.2 | 50.4 ± 6.7 | 0.0313* |
| Sex: Female | 11 | 10 | |
| Male | 13 | 10 | |
| Height (mean ± SD), m | 1.6 ± 0.1 | 1.6 ± 0.1 | 0.3629 |
| Weight (mean ± SD), kg | 68.0 ± 10.8 | 68.9 ± 17.7 | 0.5163 |
| BMI (mean ± SD), kg/m² | 26.3 ± 5.0 | 25.8 ± 7.0 | 0.4032 |
| HIV positive | 5 (21%) | 11 (55%) | |
| Antiretroviral therapy | 5 (21%) | 11 (55%) | |
| Diabetes mellitus | 0 | 0 | |
| Other medication[a] | 11 (46%) | 3 (15%) | |
| Smoker | 0 | 8 (40%) | |
| Alcohol consumption | 3 (13%) | 16 (80%) | |

*SD* standard deviation.
Statistical analysis was performed by two-sided unpaired *t*-test (data normally distributed) or non-parametric Mann–Whitney-*U*-test (data not normally distributed), statistical significance is highlighted as *=*p* < 0.05.
[a]Other medication includes anti-hypertensive, anti-depressant, and pain relief medication.
One of the 20 urban participants was underweight (BMI < 18 kg/m²).

observed in the rural Xhosa fecal microbiota (Fig. 1D). Rural fecal samples also showed high levels of OTUs related to *Treponema* (Fig. 1D), an organism that seems to be associated with a traditional rural lifestyle[16].

### Urban diet is linked to high levels of deoxycholic acid and a distinct bacteria-metabolite pattern

Global profiling of urine and fecal water from rural and urban Xhosa people was acquired and analyzed using PCA and OPLS-DA. Scores plots of PCA did not show clear clustering between the two groups and no statistically significant OPLS-DA were obtained from the two groups (Suppl. Fig. 1A, B). Targeted quantification of gut microbial metabolites revealed similar levels of the main SCFA, including butyrate, in the feces of rural and urban Xhosa people (Fig. 2A), while numbers of bacteria expressing the butyryl coenzyme A (CoA):acetate-CoA transferase (*bcoA*, involved in butyrate production) were higher in urban Xhosa fecal samples (Suppl. Fig. 1C). In contrast, levels of cholic acid (CA), deoxycholic acid (DCA) and ursodeoxycholic acid (UDCA) were higher in feces of urban Xhosa people, whereas chenodeoxycholic acid (CDCA) and lithocholic acid (LCA) were similar to the rural Xhosa cohort (Fig. 2B, C). The levels of the essential amino acids lysine and threonine, as well as serine and glutathione, were lower in fecal samples of urban Xhosa individuals (Suppl. Table 1). In contrast. the concentration of the non-dietary amino acid homocysteine was higher in urban Xhosa individuals (Suppl. Table 1), possibly reflecting their higher animal protein intake[17].

We identified a strong positive correlation between *Bacteroides*, *Dialister*, *Megasphaera*, and *Muribaculacea* with total energy intake as well as total consumption of carbohydrates, fiber, fat, protein, and several micronutrients (Fig. 2D). Conversely, *Catenibacterium*, *Christensenellaceae* and, to a minor extent, *Prevotella* showed a uniform inverse association with these dietary factors. *Prevotella*, which was enriched in the fecal microbiota of rural Xhosa individuals, had the strongest positive association with fecal SCFA levels (Fig. 2D). While this may indicate a fundamental role of *Prevotella* in saccharolytic

**Table 2 | Daily dietary intake of macro- and micro-nutrients in rural and urban Xhosa cohorts[a]**

| Daily nutrient intake | Rural | Urban | P value | Rural | Urban | p Value |
|---|---|---|---|---|---|---|
| Energy (kcal) | 2,185 ± 179 | 3,578 ± 455 | 0.0041** | **Normalized to per 1000 kcal** | | |
| Total carbohydrates (g) | 395 ± 32 | 511 ± 60 | 0.0812 | 184 ± 6 | 150 ± 7 | 0.0003*** |
| Total dietary fiber (g) | 28 ± 2 | 40 ± 5 | 0.034* | 14 ± 1 | 12 ± 1 | 0.0391* |
| Total fat (g) | 41 ± 6 | 107 ± 17 | 0.0004*** | 19 ± 2 | 28 ± 2 | 0.0004* |
| Dietary saturated fat (g) | 12 ± 2 | 31 ± 5 | 0.0013** | 5 ± 1 | 8 ± 1 | 0.005** |
| Cholesterol (mg) | 172 ± 34 | 468 ± 108 | 0.0073** | 75 ± 10 | 122 ± 20 | 0.029* |
| Total protein (g) | 63 ± 7 | 109 ± 17 | 0.0128* | 28 ± 2 | 29 ± 2 | 0.63 |
| Animal protein (g) | 25 ± 6 | 60 ± 13 | 0.0129* | 11 ± 2 | 15 ± 2 | 0.0926 |
| Plant protein (g) | 37 ± 3 | 48 ± 6 | 0.0964 | 17 ± 1 | 14 ± 1 | 0.0009*** |
| Folate (µg) | 532 ± 43 | 665 ± 95 | 0.1837 | 254 ± 15 | 197 ± 13 | 0.0086** |
| Calcium (mg) | 642 ± 75 | 809 ± 165 | 0.3348 | 299 ± 27 | 208 ± 22 | 0.0144* |
| Iron (mg) | 18 ± 2 | 27 ± 5 | 0.0582 | 8 ± 1 | 7 ± 1 | 0.0192* |
| Zinc (mg) | 11 ± 1 | 18 ± 3 | 0.0181* | 5 ± 1 | 5 ± 1 | 0.6172 |
| Vitamin A (µg) | 183 ± 67 | 625 ± 197 | 0.0273* | 69 ± 19 | 150 ± 36 | 0.0433* |
| Vitamin C (mg) | 61 ± 10 | 161 ± 69 | 0.1235 | 32 ± 6 | 37 ± 9 | 0.6269 |
| Total polyphenols (mg) | 1400 ± 212 | 994 ± 156 | 0.1503 | 830 ± 178 | 344 ± 86 | 0.026* |

[a]Values are shown as mean ± SEM (standard error of the mean). Statistical analysis for rural $n = 24$ and urban $n = 20$ samples using $t$-tests (two-sided) corrected for false discovery rate (FDR, 1%) according to the Benjamini–Hochberg method with statistical significance highlighted as *=$p < 0.05$, **=$p < 0.01$ and ***=$p < 0.001$. Source data are provided in Supplementary Data File 1.

fermentation, bacterial abundance and metabolite levels may not show linear associations. This is also exemplified by other complex polysaccharide-degrading and SCFA-producing genera: *Faecalibacterium*, *Holdemanella* and *Roseburia* were not strongly associated with fecal SCFA, whereas *Butyrivibrio* and the *Eubacterium coprostanoligenes* group were inversely associated (Fig. 2D). The fecal bile acids CA and DCA were positively associated with *Bacteroides*, *Butyrivibrio* and *Ruminococcus torques* group, whereas *Dialister* and *Succinivibrio*, both notable SCFA producers, showed an inverse correlation (Fig. 2D). A similar pattern was detected for CDCA, but not LCA, suggesting differential responses of bacteria to different classes of bile acids.

**Differences in composition but not diversity of fecal virome are linked to separate phage-bacteria-metabolite dynamics in rural and urban cohorts**

In total, 900 viral contigs (VC = viral OTUs) with lengths longer than 3 kb were identified, of which 176 were shared between the two cohorts (Fig. 3A), suggesting that a high portion remained unique to each group. The vast majority (>85%) of VCs were double-stranded DNA, and more than 4% were single-stranded DNA. The virome of rural Xhosa individuals had higher proportions of Podoviruses (3.3% vs. 0.3%) and lower levels of Siphoviruses (1.7% vs. 4.6%) and Microviruses (8.3% vs 10.5%) compared to the urban Xhosa cohort (Fig. 3B), yet these differences were not statistically significant. About 82% of the viral sequences were not taxonomically assigned. Of all VCs, 124 were found to be differentially abundant in rural and urban Xhosa individuals: phages predicted to infect *Oscillospiraceae*, *Parabacteroides*, and *Lactococcus* were more often detected in fecal samples of rural Xhosa individuals, whereas urban fecal samples contained more phages directed at *Faecalibacterium* and *Muribaculaceae* (Fig. 3C). Phages targeting *Bacteroides* were enriched in both, rural and urban, samples (Fig. 3C). With regard to viral replication cycle, more temperate phages (8.6% vs. 5%) were identified in urban fecal samples, but more crAssphages (0.2% vs. 0.02%) were found in rural Xhosa individuals. However, these differences were not statistically significant. Although there was no significant difference in α-diversity for VCs across urbanization (Suppl. Fig. 2A), these demonstrated separate clustering on Bray–Curtis dissimilarity (Fig. 3D).

Pearson's correlation of VCs that are differently abundant across urbanization and fecal metabolites or dietary nutrients revealed multiple novel associations ($r > 0.6$): VC58, a virulent phage (= a phage that is not able to lysogenize) predicted to infect *Muribaculaceae*, and the temperate phage VC80 were strongly associated with fecal butyric acid and vitamin C (Fig. 3E). VC79 was strongly linked to vitamin A levels. VC680, a virulent phage enriched in urban samples, was positively associated with cholesterol, animal protein, and zinc (Fig. 3E). Two virulent phages highly abundant in rural samples, VC139 and VC2, the latter is predicted to infect *Parabacteroides*, showed weak inverse correlation ($r = 0.2$) with dietary fiber intake, but positive correlation with the primary bile acid CA (Fig. 3E). Several VCs were positively associated ($r > 0.3$) with fecal DCA levels (VC6, VC14, VC47, VC65, VC158, VC161), suggesting a link between this secondary bile acid and phages in the gut, which requires further experimental validation.

Urban Xhosa fecal samples harbored more phages predicted to infect the phylum Bacteriodota (Fig. 3F). We were unable to assign bacterial hosts to most of the VCs in the study, with a higher abundance of unassigned phages in rural Xhosa fecal samples (Fig. 3F), highlighting the limited understanding of the gut virome in Sub-Saharan populations. At the level of VCs and bacterial genera that are both differentially abundant in rural or urban Xhosa people, a clear clustering according to dominant bacteria was detected: VC58, VC80, and VC269, all highly abundant in urban fecal samples, were positively associated with *Bacteroides* and *Fusobacterium*, but inversely linked to *Prevotella* (Suppl. Fig. 2B). VC58 and VC269 were predicted to infect Bacteroidota, suggesting active interactions between these phages and their bacterial host. VC269 was positively associated with *Bilophila*, which includes several sulfite reducers[18], and was also more abundant in urban Xhosa microbiota (Suppl. Fig. 2B).

**The composition of food and skin microbiota is different between rural and urban cohorts**

We performed exploratory microbiota analysis of the 24 h food samples collected from representative households of both Xhosa cohorts to characterize how respective food and processing conditions may contribute to the differential microbiota patterns. The microbiota of food samples collected from rural Xhosa individuals showed higher α-diversity, separate compositional clustering on Bray–Curtis similarity, a higher absolute number of bacteria counts, and higher abundances of OTUs linked to *Rosenbergiella* and *Weissella* (Fig. 4A–D), shown to be present in fruit nectar[19] or fermented foods[20], respectively.

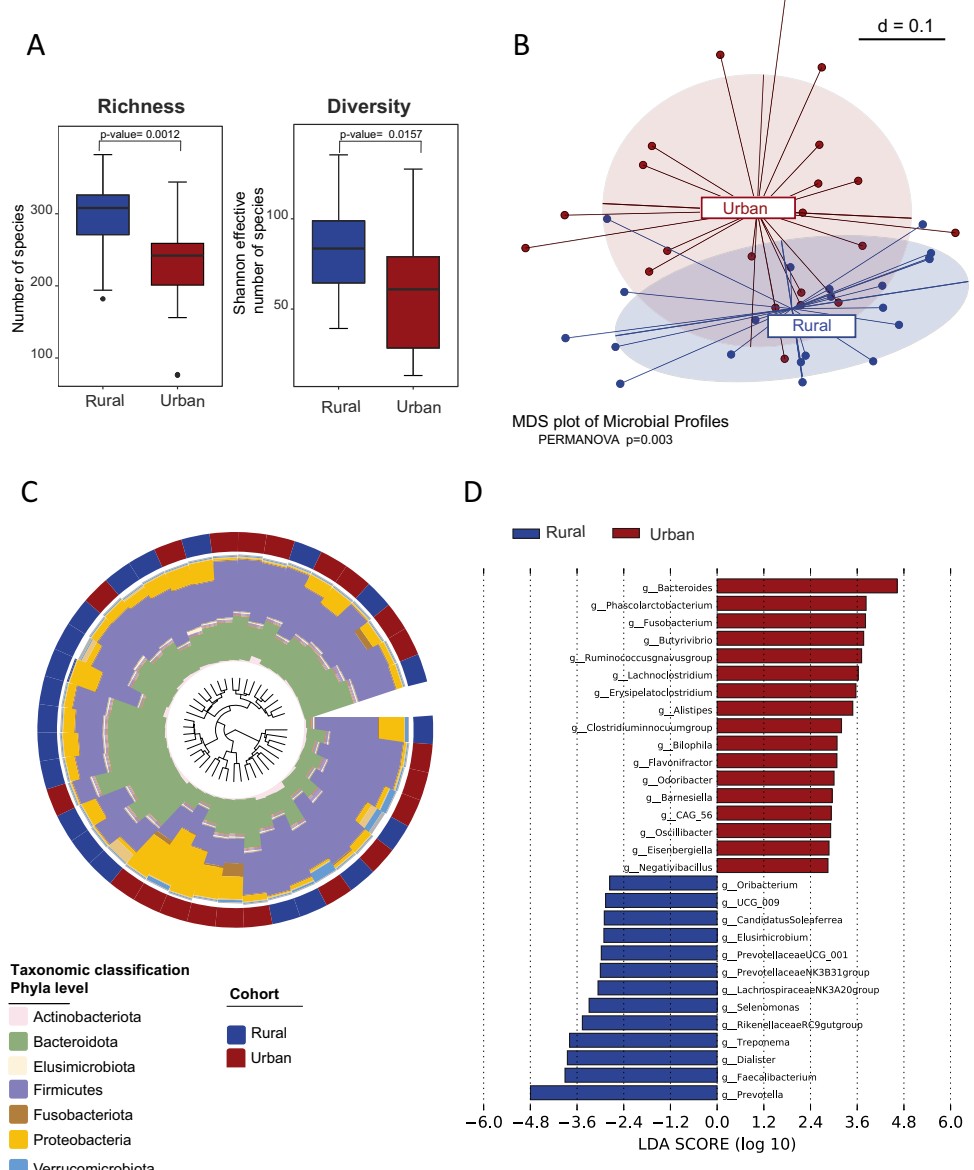

**Fig. 1 | The fecal microbiota of urban Xhosa individuals is less diverse and shows a distinct clustering and composition.** Analysis of fecal microbiota by 16 S rRNA gene sequencing from healthy rural and urban Xhosa individuals showing **A** bacterial community richness or Shannon Effective as indices for α-diversity, **B** compositional clustering at the multi-dimensional scaling (MDS) plot (PERMANOVA test $p = 0.001$), **C** cumulative relative abundance at phylum level in a phylogenetic tree and circular plot, where the outer ring indicates cohort distribution with urban (red) and rural (blue) individuals and the inner ring indicates taxonomic classification covering 100% relative abundance at phylum level for the respective sample, **D** significantly differential bacterial genera identified by linear discriminant analysis (LDA) effect size (LEfSe). Box-and-whisker plots with center as median, box from 25th to 75th percentile, and whiskers showing minima and maxima according to Tukey, Statistical analysis using two-sided Mann–Whitney-$U$-test or multivariate comparison corrected for multiple testing by the Benjamini–Hochberg method with rural $n = 21$, urban $n = 20$ samples, respectively. A value of $p < 0.05$ was considered to be statistically significant.

Regarding metabolites, the 24 h collection of the food consumed in both groups revealed higher levels of butyric acid and several other carboxylic acids such as aconitate, α-ketoglutarate, succinate and fumarate (Suppl. Table 2) in the rural Xhosa cohort.

In another exploratory microbiota analysis, hand-finger swabs of rural and urban Xhosa study participants revealed no difference in α-diversity, but distinct compositional clustering, characterized by higher levels of Cyanobacteria for rural and higher abundance of Proteobacteria in urban Xhosa individuals (Fig. 5A–C). Several low-abundance genera were more often detected in rural (e.g., *Actinomycetospora, Bacillus, Sphingomonas, Weissella*) or urban (e.g., *Acinetobacter, Jeotgalicoccus, Psychrobacter, Succinivibrio*) Xhosa individuals (Fig. 5D) (Suppl. Table 3).

## Discussion

Here, we investigate potential biome explanations for the alarming increase in NCDs such as CRC in urban South Africa by comparison to rural Xhosa individuals who continue a traditional lifestyle. We present new evidence on how diet transition across urbanization may be linked to distinct gut microbiota signatures in healthy individuals. The diet of healthy urban Xhosa individuals was characterized by higher energy, fat, cholesterol, and animal protein consumption as well as a relatively high but still <50 g/d intake of dietary fiber that was previously associated with a minimal CRC risk[2]. This was associated with a less diverse gut microbiota with bacterial compositions likely closer to Western cohorts and functionally adapted to Western diet stimuli (e.g., secondary bile acids). Despite the relatively high

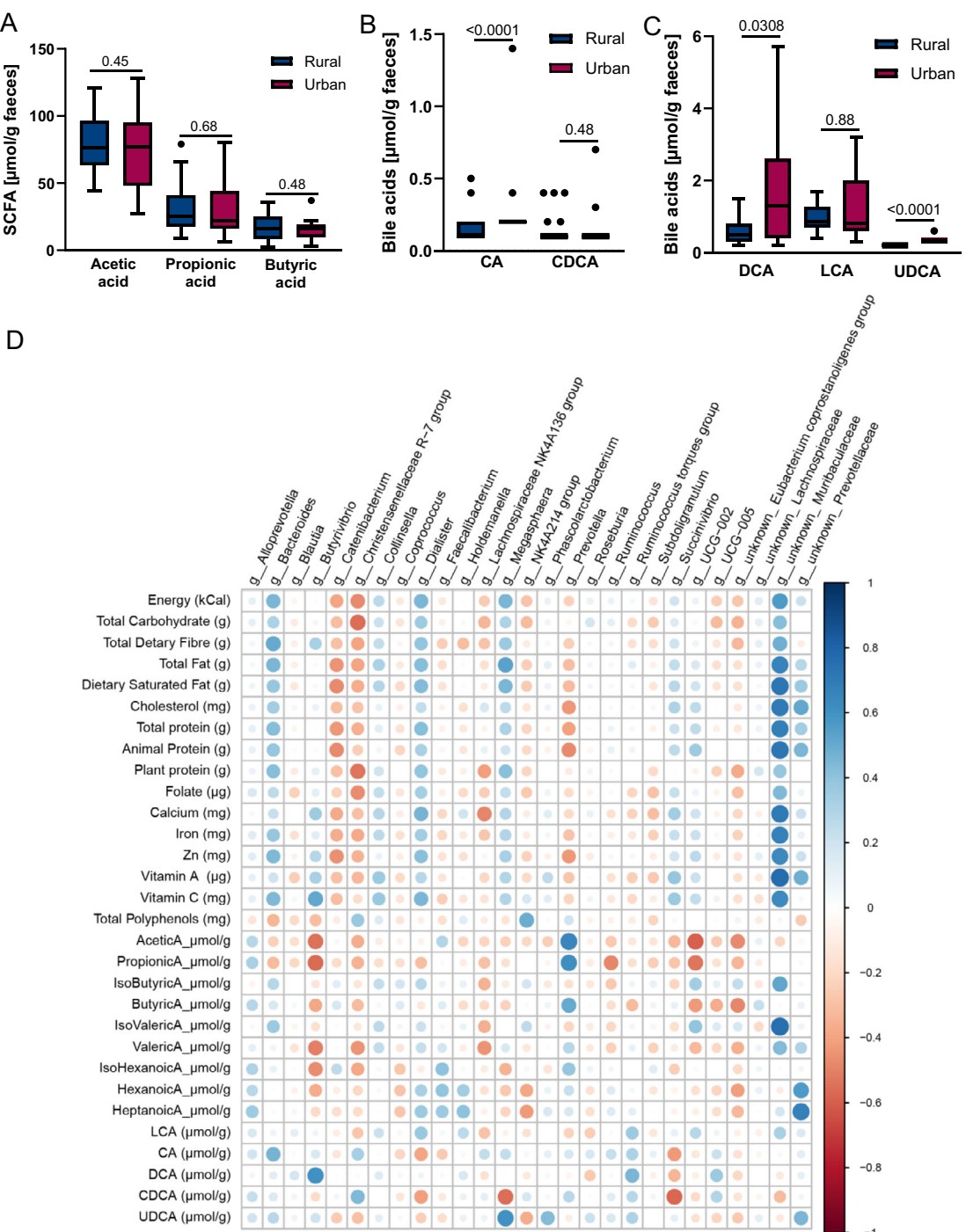

**Fig. 2 | Urban Xhosa individuals have similar SCFA but higher DCA levels compared with rural individuals, correlating with distinct microbial genera.** **A** Levels of main short-chain fatty acids detected in feces of rural and urban Xhosa individuals. Levels of **B** primary and **C** secondary bile acids detected in feces of rural and urban Xhosa individuals (CA cholic acid, CDCA chenodeoxycholic acid, DCA deoxycholic acid, LCA lithocholic acid, UDCA = ursodeoxycholic acid). Statistical analysis was performed by two-sided unpaired t-test (data normally distributed) or non-parametric Mann–Whitney-*U*-test (data not normally distributed) with rural *n* = 24 and urban *n* = 19 samples. A value of *p* < 0.05 was considered to be statistically significant. Box-and-whisker plots with whiskers minima and maxima

according to Tukey, center as median, box from 25th to 75th percentile. Source data are provided in Supplementary Data file 1. **D** Heatmap of Pearson's correlations between dietary information, fecal metabolites and microbial genera detected to be significantly different between rural and urban Xhosa cohorts using 16 S rRNA gene sequencing data. Correlations were adjusted for multiple testing using the Benjamini-Hochberg FDR. The color of the circles indicates the type of correlation (positive/negative), and the radius of the circles is proportional to the correlation. Statistical analysis was performed with sample numbers as listed for previous analyses in Table 2, Figs. 1 and 2.

amounts of dietary fiber and fecal SCFA levels detected in urban individuals, their diet resembles the pattern of Western diets and is associated with a higher number of obesity cases (*n* = 6 in urban vs. *n* = 1 in rural) (Table 1) and NCD risk-associated fecal metabolites

such as homocysteine that was previously linked to vascular diseases[21].

The fecal microbiota of the urban Xhosa cohort was enriched for bacteria and related phages that have been associated with a

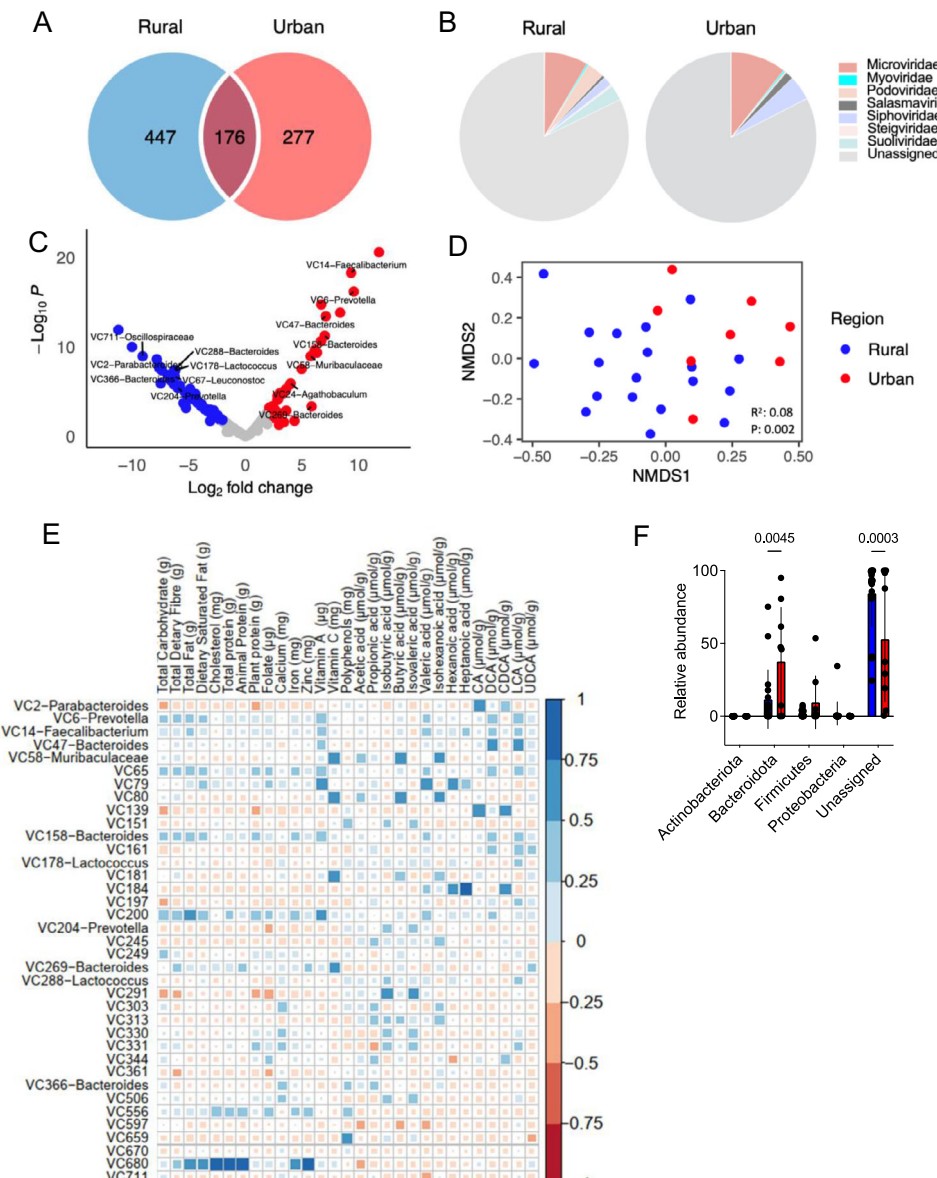

**Fig. 3 | The fecal virome of urban Xhosa individuals shows distinct composition and correlation with fecal metabolites and dietary nutrients. A** Shared and unique viral contigs (VCs) between rural ($n = 19$) and urban ($n = 9$) Xhosa individuals. **B** Proportion of VCs based on predictions of their taxa. **C** Differentially abundant VCs in rural ($n = 19$) compared to urban ($n = 9$) Xhosa fecal samples. **D** NMDS plot of Bray–Curtis dissimilarity of VCs between rural ($n = 19$) and urban ($n = 8$) Xhosa fecal samples. **E** Pearson's correlation of VCs differentially abundant in rural ($n = 19$) or urban ($n = 9$) Xhosa fecal samples and dietary nutrients or fecal metabolites, respectively. **F** Proportion of relative abundance of VCs based on predicted bacterial hosts from rural ($n = 19$) and urban ($n = 9$) Xhosa fecal samples. Statistical analysis was performed by two-way ANOVA corrected for multiple comparisons with Sidak's test. A value of $p < 0.05$ was considered to be statistically significant. When present, error bars correspond to the standard deviation of host abundances.

Western-style dietary pattern and high CRC risk in previous studies: *Bacteroides* were detected at high levels in diets rich in fat[22–25] and were more often targeted by VCs in the urban Xhosa cohort, suggesting an active interaction between this genus and its associated phages, possibly through fluctuating-selection dynamics observed in these phages[26]. *Fusobacteria* were linked to CRC in previous studies[27,28], and their phages are potential markers of CRC[29]. Sub-species of *Alistipes* showed experimental pro-inflammatory and tumor-promoting activity[30,31]. *Bilophila*, a group of sulfite-reducing bacteria associated with intestinal inflammatory conditions and adaptation of the gut microbiota to bile acid metabolism[32,33], was targeted by phages highly abundant in the urban Xhosa cohort. Phages were shown to alter the sulfur metabolism of host cells by encoding auxiliary metabolic genes[34], suggesting an active role in regulating human health through

their bacterial hosts. Urban fecal samples also contained more *Faecalibacterium* phages that are commonly found in patients with IBD[35] and may contribute to diseases by reducing the abundance of their host through kill-the-winner dynamics.

The higher fat intake of the urban Xhosa cohort is likely to increase the levels and a differential composition of fecal bile acids and to promote the abundance of gut bacteria adapted to bile acid transformation: the genus *Lachnoclostridium* includes members involved in or closely linked to bacteria performing bile acid metabolism and was enriched in Alaska Native people who have a very high CRC risk[32]. Higher levels of DCA and 7α-dehydroxylating bacteria were detected in the feces of CRC patients, in healthy cohorts facing a high CRC risk, and in urban compared with rural Zimbabweans[32,36–39]. Several phages were positively associated with fecal bile acid levels, including CA and

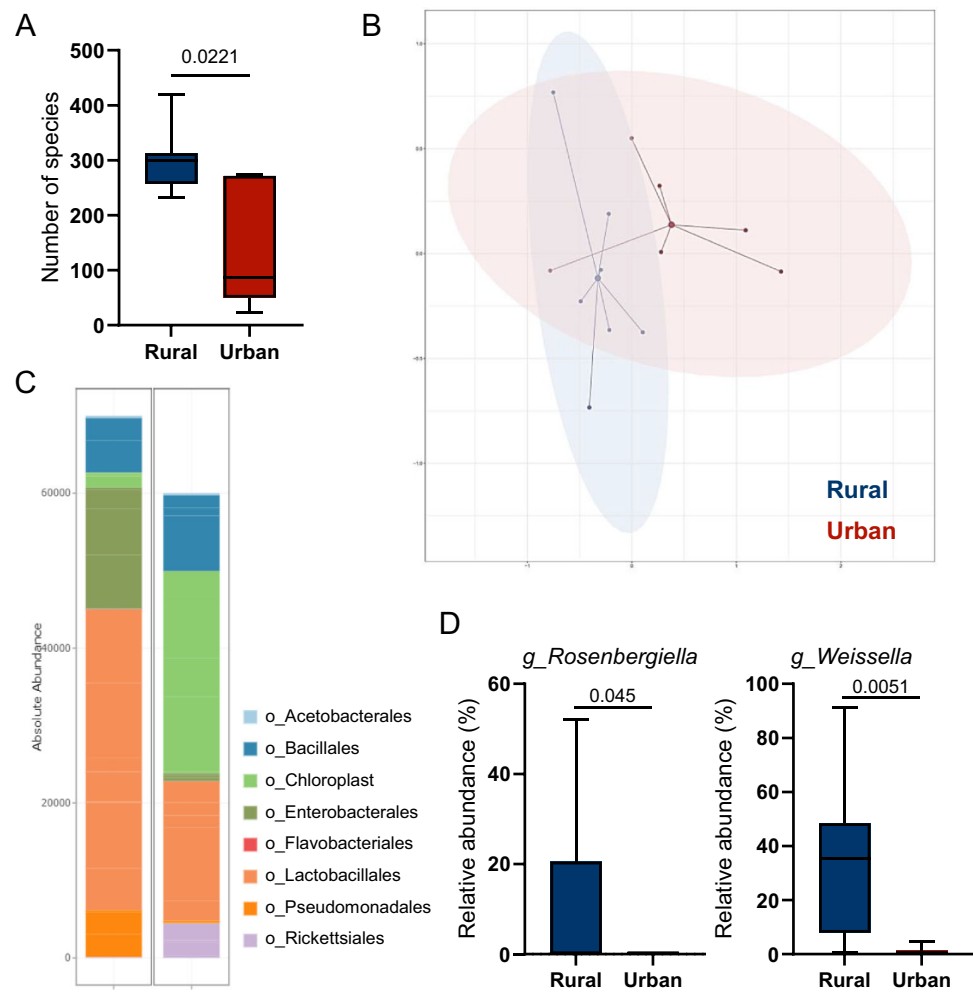

**Fig. 4 | Differences in food microbiota reflect food ingredients of rural and urban Xhosa people.** Analysis of microbiota in frozen food samples by 16S rRNA gene sequencing obtained from rural and urban Xhosa individuals showing **A** richness as an index for α-diversity, **B** Bray–Curtis dissimilarity at the multi-dimensional scaling (MDS) plot (PERMANOVA test $p = 0.004$), **C** cumulative absolute abundance at the order level, **D** significantly different relative abundance at the genus level. Statistical analysis using a two-sided Wilcoxon test corrected for multiple testing by the Benjamini–Hochberg method using normalized samples (rural $n = 7$ and urban $n = 6$). A value of $p < 0.05$ was considered to be statistically significant. Box-and-whisker plots with whiskers showing minima to maxima, center as median, box from 25th to 75th percentile.

DCA, supporting previous studies that demonstrated phage-mediated alterations of microbial bile acid transformation[40] and enrichment of a phage targeting a DCA-producing bacterium in CRC patients[29]. The higher levels of temperate phages in urban Xhosa fecal samples may be a consequence of diet transition since Western dietary compounds are potent inducers of prophages in bacteria[41] that shape the gut microbiota through host cell lysis and horizontal gene transfer[42]. In addition, the skin microbiota may be a reservoir contributing to changes in microbial setups across urbanization, especially considering different hygiene conditions or food processing approaches: higher abundance of skin-associated Proteobacteria in the urban Xhosa cohort and soil-/water-associated genera on the hands of rural individuals indicated different home environments along the urbanization gradient[43]. However, their overall very low abundances limit potent effects on the host, requiring cautious interpretation and further critical analysis (e.g., control for contamination).

Given the Western diet pattern and CRC risk-associated gut microbial composition, concomitant with retained fiber intake and high fecal SCFA levels, the urban Xhosa cohort may show a transitional state between traditional and Western diet, confirming previous results from a study in Zimbabwe[39]. This may indicate that local traditional and Western diets are interlaced in this study population, which may collectively promote beneficial effects on intestinal health across urbanization. However, functional consequences with regard to long-term CRC or NCD risk remain unclear since urban fecal samples had lower levels of saccharolytic and butyrogenic genera. Both, *Prevotella* and *Faecalibacterium*, were shown to be more abundant in fecal samples from healthy individuals compared to CRC patients in Kenya[44]. Enrichment of *Prevotella ssp.* was demonstrated to be a key characteristic of the gut microbiota in rural Sub-Saharan Africa[24,32,36,45–47]. The relative loss of *Prevotella* members in the urban fecal microbiota may be linked to the higher abundance of *Prevotella* phages in these samples and detrimental to the functional capacity to degrade plant-derived complex carbohydrates despite the high fiber consumption of the host[48,49]. This was further supported by the strong positive association of *Prevotella* with fecal SCFA in this study, but other complex polysaccharide-degrading SCFA-producers (*Holdemanella*, *Roseburia*, *Faecalibacterium*) were not strongly associated with fecal SCFA. The latter may indicate that bacteria abundance and associated metabolite levels do not follow linear associations, which is further supported by the higher *bcoA* copy but similar butyrate levels in urban compared to rural Xhosa fecal samples. Since *bcoA* gene copy

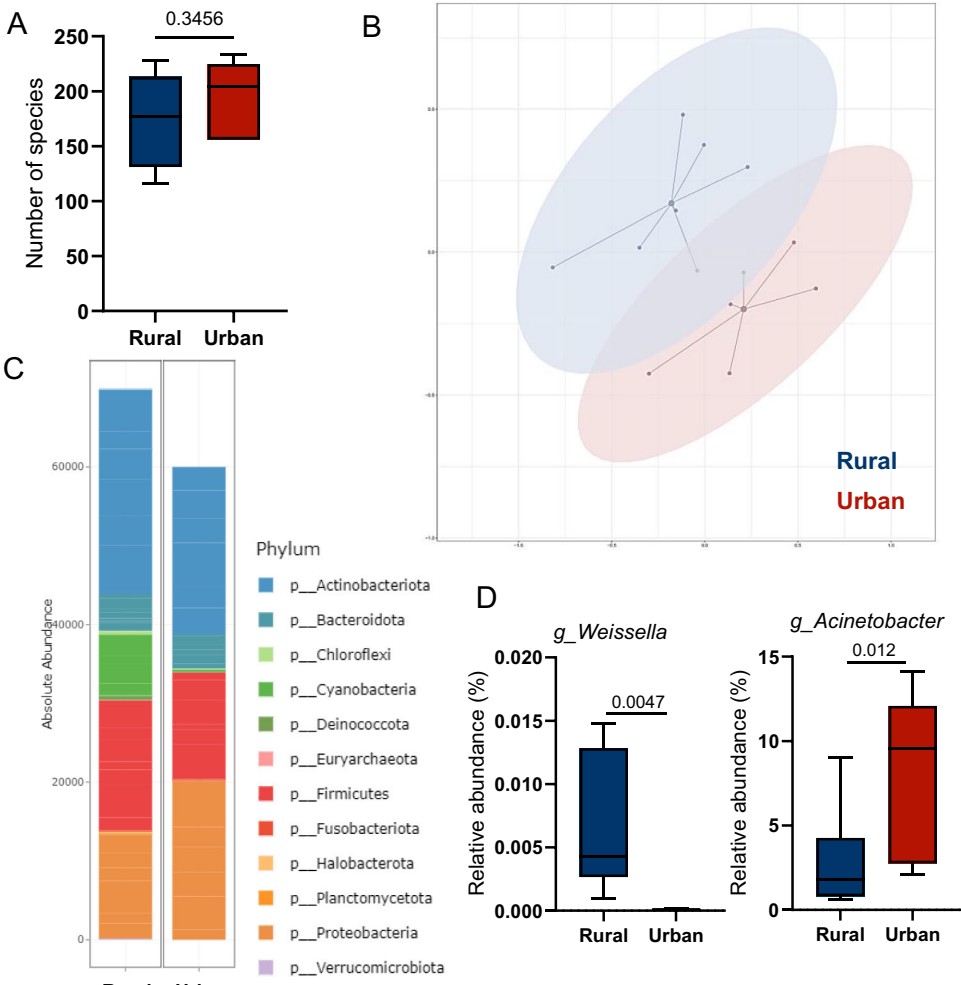

**Fig. 5 | Different bacteria detected on skin swabs of rural and urban Xhosa people.** Analysis of microbiota in skin swab samples (hands) by 16S rRNA gene sequencing obtained from rural and urban Xhosa individuals showing **A** richness as an index for α-diversity, **B** Bray–Curtis dissimilarity at the multi-dimensional scaling (MDS) plot (PERMANOVA test $p = 0.003$), **C** cumulative absolute abundance at the phylum level, **D** significantly different relative abundance at the genus level.

Statistical analysis using a two-sided Wilcoxon test corrected for multiple testing by the Benjamini–Hochberg method using normalized samples (rural $n = 7$ and urban $n = 6$). A value of $p < 0.05$ was considered to be statistically significant. Box-and-whisker plots with whiskers showing minima to maxima, center as median, box from 25th to 75th percentile.

numbers were targeted by degenerate primers designed for Western cohort studies, their higher levels detected in urban fecal samples could also reflect compositional shifts in the gut microbiota due to the transition to urban lifestyle while not covering butyrogenic bacteria that are more prevalent in rural African communities. *Treponema* was more abundant in the fecal microbiota of rural individuals and is another indicative genus for rural gut microbiota in Sub-Saharan Africa[46,50]. The genus *Phascolarctobacterium*, covering bacteria able to produce propionate and acetate, was more abundant in the urban cohort but previously reported to be reduced or lacking in Western populations and urbanized cohorts in South Africa[16,46,51]. Together with the minor differences in global fecal metabolomic profiles, these results suggest that the urban Xhosa cohort analyzed here still retains gut microbial features of rural South African communities, which may be targeted by the restoration of specific components of the traditional diet. This may include access to plant-based local food ingredients (as highlighted by the higher polyphenol intake and abundance of *Rosenbergiella* in rural Xhosa food samples) and usage of traditional food preparation such as fermentation (indicated by higher abundance of *Weissella* in rural Xhosa food samples)[19,20]. Traditional food preparation is also relevant in the context of resistant starch, which is a dietary fiber rich in traditional high carbohydrate diets of rural South

Africa, but not measured in this study as its content is variable and therefore difficult to measure with standard food composition tables[36]. This also highlights the need to characterize dietary fibers in more detail in dietary analyses, accounting for fiber heterogeneity that may affect their functionalization by the gut microbiota[4,52,53]. In this study, other factors (e.g., obesity, alcohol consumption, smoking, medication, different average age between cohorts) may impair the general extrapolation of observed biome signatures to urbanizing communities in low- or middle-income countries. Finally, the cross-sectional study layout, small sample size, and focus on two study sites may not fully reflect the dynamic processes within urbanization, whereas the lack of colonic mucosal biopsies limits the conclusions with regard to the CRC risk of study cohorts. Consistently, a recent systematic review highlighted that changes in the microbiota along the rural-urban gradient in Sub-Saharan Africa are heterogenous and multifactorial resulting in low universal patterns[54].

This study demonstrates diet- and microbiota-centered support for the hypothesis that urbanization-associated changes in lifestyle as part of economic development are the main drivers associated with NCD risk in Sub-Saharan Africa[55]. The microbial and metabolic differences of the two geographically distinct South African populations in this study warrant further investigation of the functional

consequences of diet-mediated microbiota modulation with regard to CRC and overall NCD risk. In general, more attention should be given to the quality and quantity of food consumed by inhabitants of Sub-Saharan Africa undergoing Westernization to avoid the otherwise inevitable increase in NCDs in urbanizing populations. In line with this, a focus on nutritional expertise adapted to local requirements, the characterization of undescribed African microbial diversity, including the virome, and the collaboration with local research partners and communities are integral elements to further study Westernization and NCD risk in Sub-Saharan Africa.

## Methods

### Study design and population
A cross-sectional, observational pilot study was performed in two healthy middle-aged rural or urban South African cohorts self-identifying as amaXhosa (Xhosa) with sample collection at the one-time point. The urban cohort was located in Cape Town, Western Cape Province, while the rural group was recruited in the rural Eastern Cape Province of South Africa. Cape Town is the chief city in Western Cape province and has been receiving migrant populations from within and outside the country for centuries. Most of the in-country migrants originate from rural Eastern Cape Province and have settled in the rapidly expanding township of Khayelitsha, the fastest-growing township in South Africa (population about 392,000 in 2011, according to the Department of Statistics South Africa). The study was approved by the Stellenbosch University Health Research Ethics Committee. Informed consent was obtained from all study participants. Inclusion and exclusion criteria were applied as reported previously[36], and details are provided in the supplementary information. Recruitment of healthy rural participants was based at Zithulele Hospital, Eastern Cape (see additional information in the supplementary material). Recruitment of healthy participants was organized through advertisements and visits to community leaders in public places (e.g., taxi centers and places of community worship) in Khayelitsha. In total, $n = 24$ healthy individuals were recruited in the rural and $n = 20$ healthy participants in the urban area of the study (Table 1). Our previous studies in a similar setting demonstrated that samples of this size were sufficient to characterize dietary differences of healthy populations[32,39,47]. Measurements of weights and heights were taken to calculate BMI.

### Dietary assessment
Dietary intake information of rural ($n = 24$) and urban ($n = 20$) study participants was collected at one-time point by an experienced research assistant specifically trained in dietary intake assessment. A South African country-specific, validated semi-quantitative food frequency questionnaire (QFFQ) was employed, developed for the Prospective Urban and Rural Epidemiology study[56] together with a 24-h recall questionnaire. Dietary intake data were quantified in a non-blinded fashion using dietary coding sheets, Food Quantities Manual, and Food Composition Tables developed by the South African Medical Research Council[57,58]. The amounts of nutrients per day and nutrients per specific food items consumed by the two study populations were estimated by computer analysis. Polyphenol values were quantified using the online database Phenol-Explorer (http://phenol-explorer.eu). Source data are provided in Supplementary Data File 1.

### Sample collection
Collection tubes with and without DNA stabilization buffer (Invitek Molecular Inc., Germany) for the collection of fecal and urinary (collected in the morning) samples were prepared to facilitate collection in home environments. The sampling took place from November 2019 to February 2020. Duplicates of all meals and drinks consumed over 24 h were collected as "total food intake" from rural ($n = 7$) and urban ($n = 6$)

Xhosa individuals, kept cool until complete, and transported to the laboratory for homogenization and sampling. Hand-finger swabs were taken from the same study participants (rural $n = 7$, urban $n = 6$) using a DNA stabilization buffer collection tube (DNA Genotek Inc., Canada). Samples were transported on ice to the laboratory, where they were aliquoted and frozen at −80 °C as soon as possible, generally within 6 h. The samples collected in rural Eastern Cape were first frozen at −20 °C at the clinic and then air-couriered on dry ice to the laboratory for further processing.

### Quantification of fecal SCFA by gas chromatography–mass spectrometry (GC–MS)
Fecal SCFA was analyzed using a stringently validated and simplified method for the quantitation of SCFA in human stool using gas chromatography–mass spectrometry[59]. Calibration curves of each SCFA were prepared by extracting aliquots of the Sigma-Aldrich VFAM Certified Reference Material equivalent to 0.05, 12.5, 25.0, 37.5, and 50.0 μmol/g using the internal standard extraction solution following the procedure described.

### Quantification of fecal bile acids by GC–MS
Selected fecal bile acids (cholic acid (CA), chenodeoxycholic acid (CDCA), deoxycholic acid (DCA), lithocholic acid (LCA), ursodeoxycholic acid (UDCA)) were quantified by GC-MS operating in selected ion monitoring mode using $D_5$-chenodeoxycholic acid as an internal standard. The LDR was 0.25–5.00 μmol/g feces. The LOQ and detection limit was 0.25 μmol/g feces. Both the coefficients of variation and the relative recoveries were less than 5% and 90–110%, respectively, for most compounds at most levels.

### $^1$H NMR spectroscopy-based metabolic profiling of urine and fecal water
Untargeted profiling of urinary and fecal metabolites was performed as described previously[32,36] and is detailed in the supplementary information.

### Quantitation of amino acids by ultra-high-performance liquid chromatography with mass spectrometry (UPLC–MS/MS)
The quantification of amino acids in feces was performed using LC-MS/MS, with details provided in the supplementary information.

### Food metabolomic analysis by LC–MS/MS
Targeted quantitative metabolomics was applied to analyze food samples using a combination of direct injection mass spectrometry with a reverse-phase LC–MS/MS custom assay. This custom assay, in combination with an ABSciex 4000 QTrap (Applied Biosystems/MDS Sciex, MA, USA) mass spectrometer, was used for the targeted identification and quantification of up to 207 different endogenous metabolites including amino acids, acylcarnitines, biogenic amines & derivatives, uremic toxins, glycerophospholipids, sphingolipids and sugars[60], details are provided in the supplementary information.

### Bacteriome analysis and quantification of microbial bcoA copy numbers
For bacteriome analysis of fecal samples stored in DNA stabilization buffer (rural $n = 24$, urban = 20) and frozen food (rural $n = 7$, urban $n = 6$) samples, 16 S rRNA gene sequencing analysis was conducted as described previously[61] with few changes. Briefly, the DNA of the samples was isolated from 600 μl of the sample-stabilizer mixture after bead beating as described but using the MaxWell (Promega) with the Maxwell® RSC Fecal Microbiome DNA Kit. Isolated DNA was used in a 2-step PCR, first using specific primer 341 F (CCT ACG GGN GGC WGC AG) and 785 R (GAC TAC HVG GGT ATC TAA TCC). Both specific primers had adapters attached, which were used in the second PCR to

attach sample-specific barcodes and the P5 and P7 adapters for sequencing. The 16 S rRNA gene sequencing analysis of hand-finger swab samples (rural $n = 7$, urban $n = 6$) was performed as previously described[32]. After sequencing, the raw data were processed using the functionality of the IMNGS database[62], details are provided in the supplementary information. For quantification of the butyryl coenzyme A (CoA):acetate-CoA transferase (*bcoA*) that is involved in butyrate production, extracted microbial DNA was analyzed as described before using the following degenerate primers (*bcoA* F: GCI GAI CAT TTC ACI TGG AAY WSI TGG CAY ATG, *bcoA* R: CCT GCC TTT GCA ATR TCI ACR AAN GC)[32].

## Fecal virome analysis

Virus-like particles (VLPs) were isolated and processed from 28 fecal samples (rural $n = 19$, urban $n = 9$) as described previously[63]. One sample from the urban cohort was excluded as an outliner in the NMDS plot. Viral sequencing data were processed using ViroProfiler[64]. The methods performed for isolating, sequencing, and characterizing the fecal viruses are detailed in the supplementary information.

## Statistical analyses

Statistical analysis of data sets on dietary parameters, fecal SCFA, bile acids, and *bcoA* gene copy numbers was performed by two-tailed unpaired t-test or non-parametric Mann–Whitney-*U*-test (depending on normal distribution of data) using Prism v.9.0 software (GraphPad Software Inc., CA, USA). Dietary data was FDR (1%) corrected according to the Benjamini–Hochberg method. Statistical analysis of bacteriome data was conducted using the Rhea pipeline and NAMCO microbiome explorer[65,66]. Briefly, adjustment for multiple comparisons was done by the Benjamini–Hochberg correction, and depending on the type of comparison, different statistical tests were applied as described, including Kruskal–Wallis Rank Sum, Fisher's Exact, and Wilcoxon Rank Sum test. Microbial β-diversity was determined by fitting models with 16 S profiles as distance-based responses using PERMANOVA and visualized by clustering on an MDS plot. For statistical analysis of virome data, Prism v.9.0 software was used to run two-way ANOVA and Sidak's multiple comparisons test, and the remaining analyses were conducted in R v.4.2.1 (R Core Team, 2022).

## Reporting summary

Further information on research design is available in the Nature Portfolio Reporting Summary linked to this article.

## Data availability

The bacteriome and virome sequencing data generated in this study have been deposited in the NCBI BioProject database under accession code PRJNA1066974 and in the ENA under accession code PRJEB67451, respectively. Dietary and targeted metabolomics data are available in a Supplementary Data File.

## Code availability

Software programs used to analyze the data are either freely or commercially available. All other data relevant to the study are included in the article. Additional data are available on request.

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

## Acknowledgements

The authors would like to thank Jinling Xue for her assistance in extracting the viral DNA. S.J.D.O. received funding from the South African Research Foundation (SRG210331591524), the Stellenbosch University Rectors Strategic Research Fund, and the NIH (R01 CA204403).

S.O. received funding from the German Federal Ministry of Education and Research (01KA2103).

## Author contributions

S.J.D.O. designed, acquired funding, conducted, and edited the study. S.O., M.C.R., and M.K.M. prepared the original draft of the paper. M.C.R, L.T.N., A.S.W., S.F. helped with administration and regulatory committees and helped conduct the cross-sector study. S.K.J. and R.S. helped with the dietary analyses. M.K.M. was responsible for the virome analyses and evaluation. B.L.E. performed the targeted metabolite analyses. M.V.-G., A.M., G.B., J.R., D.M.-J., D.C.L., J.Z., Q.C., and M.R.P. helped with the laboratory sample analysis. A.M., S.O., K.N., and B.M. performed the bacteriome analysis. J.V.L., D.W., and R.M. performed the metabolomic analyses. B.G. organized the recruitment of rural volunteers. S.O., D.H., J.V.L., and L.D. supervised and organized the analysis of samples.

## Competing interests

The authors declare no competing interests.

## Additional information

M.C. Ramaboli [1,15], S. Ocvirk[2,3,4,15], M. Khan Mirzaei [5,6], B. L. Eberhart[2], M. Valdivia-Garcia[7], A. Metwaly [8], K. Neuhaus [9], G. Barker[7], J. Ru[5,6], L. T. Nesengani[10], D. Mahdi-Joest [3], A. S. Wilson[2], S. K. Joni[11], D. C. Layman[11], J. Zheng[12], R. Mandal[12], Q. Chen[7], M. R. Perez [2], S. Fortuin[1], B. Gaunt[13], D. Wishart[12], B. Methé[14], D. Haller [4,8], J. V. Li [7], L. Deng [5,6], R. Swart [11] & S. J. D. O'Keefe[1,2] ✉

[1]African Microbiome Institute, Department of Biomedical Sciences, Faculty of Medicine and Health Sciences, Stellenbosch University, Cape Town, South Africa. [2]Division of Gastroenterology, Hepatology and Nutrition, Department of Medicine, University of Pittsburgh, Pittsburgh, PA, USA. [3]Intestinal Microbiology Research Group, German Institute of Human Nutrition, Potsdam, Germany. [4]ZIEL - Institute for Food and Health, Technical University of Munich, Freising, Germany. [5]Institute of Virology, Helmholtz Centre Munich - German Research Centre for Environmental Health, Neuherberg, Germany. [6]Chair of Microbial Disease Prevention, School of Life Sciences, Technical University of Munich, Freising, Germany. [7]Section of Nutrition, Department of Metabolism, Digestion and Reproduction, Faculty of Medicine, Imperial College London, London, UK. [8]Chair of Nutrition and Immunology, TUM School of Life Sciences, Technical University of Munich, Freising, Germany. [9]Core Facility Microbiome, ZIEL - Institute for Food and Health, Technical University of Munich, Freising, Germany. [10]Department of Agriculture and Animal Health, University of South Africa, Pretoria, South Africa. [11]Department of Nutrition and Dietetics, School of Public Health, University of the Western Cape, Cape Town, South Africa. [12]The Metabolomics Innovation Centre & Department of Biological Sciences, University of Alberta, Edmonton, Alberta, Canada. [13]Zithulele Hospital, Mqanduli District, Mqanduli, Eastern Cape Province, South Africa. [14]Center for Medicine and the Microbiome, Department of Medicine, University of Pittsburgh, Pittsburgh, PA, USA. [15]These authors contributed equally: M. C. Ramaboli, S. Ocvirk. ✉e-mail: sjokeefe@pitt.edu

