## [Peer Review File · Nature Communications]

REVIEWER COMMENTS

Reviewer #1 (Remarks to the Author):

In their article, “Diet changes due to urbanization in South Africa are linked to microbiome and metabolome signatures of westernization and increased colorectal cancer risk” Ramaboli MC et al report on a cohort of 24 and 21 rural and urban residents and make associations between the subjects’ diet, their bacterial microbiome, metabolome, and virome. The breadth of analyses makes this a very interesting cohort where new associations are possible between dietary intake and microbiome composition and function. The cross-sectional design and the combination of multiple metagenomic and metabolomic analyses permits testing of associations between diet, microbiota, and microbial functions. However, the dataset is not sufficiently representative to support the authors’ claims that they are ‘signatures of westernization’ and the manuscript provides no data to support the claim of these microbiota changes increasing risk for colorectal cancer (or other non-communicable diseases). Shortcomings are that this sample size of 45 individuals is not representative of South Africa, a rural setting and/or an urban setting. Additionally, the cross-sectional design with one time point in a diverse set of individuals makes it unlikely that diet analysis is sufficiently representative of dietary intake and biosampling sufficiently representative of microbiome composition and function. Finally, the authors generalize their findings to “westernization and increased colorectal cancer risk” when their analyses do not address these claims or risk – e.g. diets are not compared to “western diets”(questionable if this exists) and patient risk factors for NCDs and/or colorectal cancer are not described. Finally, the authors include no discussion of the validity and limitations of their study results. Suggest authors temper their claims and concentrate on the unique overlapping set of cross-sectional data in these 45 individuals without extrapolation to metabolic and CRC risks that they do not prospectively measure.

Results

Suggest that authors start their results with a description of the included cohort, noting sample size and Table 1.

How do the authors think that 24 individuals is representative of ‘urban’ vs ‘rural’. Rather big generalizations from such a small cohort of individuals, particularly given the diversity of eating patterns over space and times in settings like these. Also address the limitation of the cross-sectional design. Clarify that dietary analyses and sampling only occurred at one time point.

Did the authors collect data on demographic variables like co-morbidities, medications, antibiotic use that could be potential confounders of the analysis.

Table 2 – confirm p values are corrected for FDR

Figure 1 – why is the sample size here 21 when Table 1 describes 24 and 21 individuals?

Figure 1A – correct y axis for Shannon diversity

Figure 1B – include axis for MDS plot

Lines 106 – authors have numerous comments about the function of genera that are significant in their analysis. However their analyses did not look at the functions of these bacteria. Suggest to remove these comments and move any description of potential functions and comparison to previously published literature to the discussion. Same for line 123 in comments about homocysteine

Metabolomics

Line 126-138 – authors associate abundance of bacteria to bacterial metabolic profiles. While suggestive that there are causal links, there is not necessarily a linear relationship between genera abundance and bacterial function, please address and modify.

Virome

Include sample size for viral analyses – looks like there were only 8 urban participants included in the analysis? (fig 3D)

The authors report on RNA viruses, but RNA viral analysis not described in supplementary material (line 145). If no RNA analysis was done, make this explicit in results and methodology

Describe detection of any eukaryotic viruses.

Authors used IPHoP to predict viral contig host assignment and BACPHLIP for replication cycles. Host assignment and description of replication is often poor. Can authors provide likelihood of accurate assignment to inform readers of the strengths of these associations/claims.

Lines 153-4 add p-values to comparisons

Line 165 remove 'major'

Figure 3E, again, questionable validity of associating phage data with all dietary and metabolomic parameters. Particularly given lack of knowledge of temperate/lytic activity. Consider removing or moderating language describing associations to acknowledge possible shortcomings.

Authors use the term 'virulent' for several phages – how did they make this classification? Suggest to either define or remove

Figure 3F – can the authors provide a better rationale for this comparison of phage host assignments between urban and rural subjects given the huge percentage of undefined viral contigs in combination with the expected accuracy of the phage-bacterial host assignments.

Composition of food and skin microbiota

Line 177, modify text “is linked to urbanization of lifestyle” to be more reflective of strength of associations. e.g. there are differences between two cohorts residing in urban and rural settings.

It is unclear how the microbiota of skin plays into the central research questions of this article. Why were the authors interested in the skin microbiota, and what hypotheses were they testing with their analysis?

Figures – suggest to use one denotation for groups – either HE/HK or urban/rural

Discussions

Include a paragraph on the limitations of this study.

The authors show interesting associations between rural urban residence, dietary intake and microbiome. However their sample size is small, they do not look at risk of colorectal disease or NCD, and should remove claims of associations with these risks from the discussion.

Additionally, “westernized” diet is a misleading term – differences in dietary intake varies enormously between individuals and is likely highly heterogenous across urban and rural settings, can the authors address inter-individual vs intergroup differences in their dietary analyses?

Line 205: “despite the relatively high amounts of dietary fiber... their ‘Westernization of diet’ is likely being effective in promoting NCD risk, reflected by higher number of obesity cases” An alternative hypothesis is that there are differences in the microbiota because of gross baseline differences in the cohort, such as obesity, energy expenditure, and environmental exposures that are broader than diet alone. Please address.

Line 209 – suggest to moderate this claim – authors don’t look at CRC risk. Suggest to remove onward suggestions that a ‘westernized diet’ in this cohort automatically gives a higher risk of CRC.

Line 212 – this sentence suggests Fusobacteria in this cohort were linked to CRC incidence, which is not the case, suggest to revise.

Methods

Dietary assessment – one time? How representative? Blinding?

Line 309 – what is meant by ‘collections of food intake’

Clarify number of samples collected per individual subject

Urinary samples not described

Line 332 – title describes urine metabolic profiling, but not in description of activities

Line 363 – RNA and DNA or only DNA?

How did authors deal with multiple sampling from one fecal sample? What were the numbers of freeze thaw cycles?

Reviewer #3 (Remarks to the Author):

The study by Ramaboli MC et al. presents interesting findings that focus on linking the dietary intake and the fecal microbiome (bacteriome and virome)/metabolome profiles since multiple diseases including colorectal cancer (CRC) incidence associated with westernized high-fat diets by comparing healthy rural and urban Xhosa people in South Africa. It also supports the hypothesis that urbanization is associated with changes in lifestyle as part of economic development and is main drivers of non-communicable disease risk in sub-Saharan Africa. The manuscript is well-written, with clear descriptions and interpretations of the results.

To improve the quality of the manuscript, the following issues need to be addressed.

1. Line 110: Prevotella and Faecalibacterium were the main microbiotas highly abundant in rural individuals compared to urban individuals on a Westernized diet. A recent study showed a high abundance of Prevotella copri and Faecalibacterium prausnitzii in Kenyan healthy volunteers compared to CRC patients. This report supports the current findings (Obuya S, J Gastrointest Oncol. 2022;13(5):2282-2292. doi:10.21037/jgo-22-116).
2. Line 110& 218: In sufficient rationale/discussion related to findings that urban fecal samples also contained more Faecalibacterium phages that are commonly found in patients with IBD. This makes Faecalibacterium low abundant in urban compared to rural Xhosa fecal microbiota.
3. Line 114: Bile acid secretion differs according to the nature of the consumed food that day. Was the fecal sample collected once and analyzed for the microbiome and the metabolite profiling?
4. Line 120: It is believed that levels of bile acids (cholic acid, deoxycholic acid, and ursodeoxycholic acid) were higher in the feces of urban Xhosa people because they consume high-fat diets.
5. Line 210: If bacteriophages infect and kill bacteria and if Bacteroides were detected at high levels in diets rich in fat. In this case, bacteriodes-targeting viral contigs should be less in the urban Xhosa cohort eating high-fat diets. Please explain.

6. In methodology: Please provide the inclusion and exclusion criteria since the current study was done in CRC patients. It is also better to mention the alcohol consumption and the smoking status of individuals, if it is available since previous studies showed that they affect the microbiome.

7. If it is possible, a confirmative qPCR for the main species found in the study would be an excellent validation.

Changes due to Urbanization in South Africa are Linked to Microbiome and Metabolome Signatures of Westernisation and Increased Colorectal Cancer Risk

Reviewer comments

August 2023

Introduction

While the importance of epidemiological transition in Africa – including lifestyle and diet – to health is understood, the topic remains under-studied. The microbiome is an important mediator and biomarker of the impact of change of diet, and so it is particularly useful to have studies which can describe how urbanisation impacts on the microbiome. Dysbiosis in the microbiome is linked to increased risk of colorectal cancer.

This paper studies compares the microbiome and metabolome of ≈ 40 amaXhosa, half urban, half rural.

Analysis of food intake is complex because the urban participants had a calorie intake 60% greater than the rural participants. So, urban participants had significantly more fibre than rural participants, but as a proportion of the diet, rural participants had more fibre (though this was not statistically significant). Urban dwellers has much more fat in total (highly statistically significant) and in proportion of diet (p-value, 0.06-0.08). Urban dwellers have a (non-statistically significant) greater total polyphenol consumption but a very statistically smaller normalised polyphenol consumption.

Analysis of the microbiome shows significantly smaller α -diversity in the urban dwellers and suggestive clustering when using β -diversity as a lens. Interesting and expected differences between key taxa can be found. Study of the metabolome does not show distinct clustering though there are significant differences in key metabolites.

An analysis of the difference in the virome also reveals differences between the two groups. I thought particularly interesting that in the urban group, there were more viruses that are associated with bacteria often characterised as healthy.

Finally, there was an analysis of food and skin microbiome – interesting but only briefly reported.

Evaluation

1. The strengths of this research is that they have
 - An urban and rural groups in the same country which are matched with respect to ancestry;
 - Good characterisation of food intake (ffq for almost all, and detailed analysis of a sub-sample);

- Microbiome and metabolome analysis.
2. The interlinked analysis of the microbiome, metabolome and food is interesting.
 3. The authors don't identify limitations of the study. However, I think a weakness is the relatively small sample size and the insufficiently described groups. The authors say the sample size is in their experience big enough to detect differences between the groups. On the whole I buy this argument, but I think that a more serious problem is that in a complex society is whether in this respect the individuals who volunteered are representative of the population as a whole. See some detailed comment below.
 4. While at a high level the methodology seems sound, I would like to see more detail. For example, the bioinformatics analysis is not reproducible by anyone else – I could do a *similar* thing but not exactly what they did
 - For LDA the authors say “LDA ... was calculated as before ... [7]”. I assume it's reference 7 of the supplementary since reference 7 of the main text doesn't seem appropriate. Reference 7 of the supplementary is a general introduction to LEfSe (I don't think authors in common so not sure what “as before” means).
 - Ideally, the authors should describe exactly which scripts they used (and versions) with what parameters. If they have their own scripts they should be available. Which versions of the database were used.
 5. I did not see a commitment to making the data available (I searched as well). From what I understand there is no personally identifying data here and so there is not reason why the data should not be available in a public. The impact of this paper will be affected by whether the data is available. I understand this is not required for review, so I assume the authors will address this should the paper be accepted.
 6. In the discussion, much is correctly made about the transitional nature of the urban diet. To what extent is this transitional diet reflected in the microbiome – how does the microbiome differ with that found in other urbanised settings and what does this say about NCD risk and CRC risk in particular? There are some pointers on page 5 but more perhaps could be made.
 7. Overall, I think the paper and the data produced make a valuable contribution to understanding epidemiological transition, and will be of interest to other researchers in the area.

Other comments

8. The supplementary material should describe the cohorts in detail (you could fit one 1 page a table with urban participants on the left and rural on the right, showing age and BMI and sorted by sex and BMI).

The authors give standard error of the mean – I think that standard deviation would be more helpful to show range. I am doubtful given that you don't truly have random sampling that that shown standard error of mean is useful. To say that the average age of the rural cohort is 57 with a standard error of the mean being 2 when you have explicitly set limits on the age doesn't really tell us anything about the population

sample. And given the population age profile a truly random sample would likely have a younger mean, I think.

The numbers don't seem to add up – in particular were participants excluded with > 35 BMI.

- Line 288 says a BMI ≤ 35 is an inclusion criterion.
 - Line 294 says 24 rural and 21 urban participants were selected.
 - Line 297 says 24 rural and 20 participants had dietary data assessed through questionnaire.
 - Line 388 says 21 samples were used in both urban and rural – was this a decision to have an equal number or did some fail QC.
 - Table 1 says that 1 of rural and 5 of the urban participants had a BMI > 35 .
 - Table 2 reports on all 44 samples for which dietary information is given.
 - Table 1 lists 10 male and 10 female participants – did one participant not self-identify as male or female?
9. line 76-78: I don't think it is true that NCDs are still relatively uncommon – even in some rural populations though obviously what counts as rural varies. Anyway, I think you need a citation to a recent (pref after 2021) high-quality paper that talks of the incidence of NCDs in *South Africa* here.
 10. line 93: what does “distribution was within normal limits”. Do you mean that it is within the range normally found in western diets?
 11. line 99: the p -value for cholesterol in Table 2 is wrong
 12. line 105/106 and throughout. I understand your use of HK and HE but I don't think they're the most obvious – rather use R and U for rural and urban.
 13. line 106: The claim that 1C makes is not obvious to me – caption needs to be longer
 14. line 145-147: these differences look very different. Did you subject them to a formal statistical test?
 15. Line 157: I think better: “, which are...,” to “that are ...” (no commas)
 16. Line 258-260: This may be a point worth making before. Although you have good quality food data, the high level descriptors can be misleading. Not all fibre is the same, not all fat is the same
 17. Line 282: The word ‘homeland’ should be avoided in a South African context – too loaded a term. I am also concerned about the use of the word ethnicity since it is a complex term – I think better would be “self-identify as amaXhosa”.
 18. Line 283: centuries
 19. Line 285: I think the population claim needs a good reference. Given the population of the whole province is about 7m, I find 2m for Khaylitsha as surprising. I see the number is highly contested in the Wikipedia article on Khayalitsha.

20. Line 286-288: I think make it easier on the reader and put the inclusion criteria in the supplementary material. I had a look at reference 15. Did you rely on self-reporting for exclusion for diabetes?
21. Line 289 – A brief description of the district may be useful. In the context of SA’s rapid transition there is considerable diversity in rural areas. To what extent do people in the district buy or produce food?
22. Line 308: what kits were used for the gut microbiome collection?
23. Lines 368-378: I think this discussion is at the right level of abstraction for the main body. But I think more detail is needed in the supplementary, otherwise this is not reproducible.
24. In Table 1, I don’t think using Xhosas as a plural is correct. Grammatically this should be adjectival describing “Sample Population” and so singular.

REVIEWER COMMENTS

Reviewer #1 (Remarks to the Author):

In their article, “Diet changes due to urbanization in South Africa are linked to microbiome and metabolome signatures of westernization and increased colorectal cancer risk” Ramaboli MC et al report on a cohort of 24 and 21 rural and urban residents and make associations between the subjects’ diet, their bacterial microbiome, metabolome, and virome. The breadth of analyses makes this a very interesting cohort where new associations are possible between dietary intake and microbiome composition and function. The cross-sectional design and the combination of multiple metagenomic and metabolomic analyses permits testing of associations between diet, microbiota, and microbial functions. However, the dataset is not sufficiently representative to support the authors’ claims that they are ‘signatures of westernization’ and the manuscript provides no data to support the claim of these microbiota changes increasing risk for colorectal cancer (or other non-communicable diseases). Shortcomings are that this sample size of 45 individuals is not representative of South Africa, a rural setting and/or an urban setting. Additionally, the cross-sectional design with one time point in a diverse set of individuals makes it unlikely that diet analysis is sufficiently representative of dietary intake and biosampling sufficiently representative of microbiome composition and function. Finally, the authors generalize their findings to “westernization and increased colorectal cancer risk” when their analyses do not address these claims or risk – e.g. diets are not compared to “western diets” (questionable if this exists) and patient risk factors for NCDs and/or colorectal cancer are not described. Finally, the authors include no discussion of the validity and limitations of their study results. Suggest authors temper their claims and concentrate on the unique overlapping set of cross-sectional data in these 45 individuals without extrapolation to metabolic and CRC risks that they do not prospectively measure.

Results

Suggest that authors start their results with a description of the included cohort, noting sample size and Table 1.

Thank you, we have added a brief description of the cohort incl. sample size to the results section.

How do the authors think that 24 individuals is representative of ‘urban’ vs ‘rural’. Rather big generalizations from such a small cohort of individuals, particularly given the diversity of eating patterns over space and times in settings like these. Also address the limitation of the cross-sectional design. Clarify that dietary analyses and sampling only occurred at one time point.

Thank you for the comment. We agree that the number of 44 individuals in total is not representative for the urban or rural population in South Africa and having one time point of dietary and sample analysis does not go beyond reflection of a snapshot. However, there are factors indicating that these relatively small but well-characterized cohorts may be used to represent generalized rural-urban nutrition transition patterns in South Africa: given the low dietary diversity in South Africa with rural areas having lowest scores (Shisana et al. South African National Health and Nutrition Examination Survey (SANHANES-1) HSRC Press; Cape Town, South Africa: 2013; Labadarios D. et al. How diverse is the diet of adult South Africans? Nutr. J. 2011;10:33), even small cohorts may cover the dietary transition to an urban diet pattern that is characterized by higher dietary diversity. Similar overall differences in diet between rural and urban Sub-Saharan cohorts were also shown in our previous studies (Ou et al. AJCN 2012; O’Keefe et al. Nat Comm 2015; Katsidzira et al., Nutr Canc 2019). We have added a paragraph to critically discuss the limitations of the study. We have also added a statement in the respective methods sections that dietary analysis and sample collection were performed at one time point.

Did the authors collect data on demographic variables like co-morbidities, medications, antibiotic use that could be potential confounders of the analysis.

We have collected data on demographic variables and included these in the revised table 1. Of note, HIV, smoking and alcohol consumption status differ substantially between rural and urban cohorts, and may therefore be confounding factors affecting the results of the analysis. In addition, self-reported medication could affect microbiota composition and function. A sentence describing a limitation of the study was added to the discussion.

Table 2 – confirm p values are corrected for FDR

The revised table 2 shows all p-values corrected for FDR.

Figure 1 – why is the sample size here 21 when Table 1 describes 24 and 21 individuals?

Three samples were excluded following quality control used for removing low quality 16S rRNA gene sequencing samples (due to values below cutoff value of 10.000 reads). In addition, there was a wrong number added for the urban cohort in table 1 - there were n=20 participants from the urban cohort in total. This was corrected in table 1.

Figure 1A – correct y axis for Shannon diversity

In Figure 1A, we show community diversity as “effective number of species” by converting Shannon index into an effective number of species following this formula e^H (exponential of Shannon entropy index) according to^{1,2}. To address this comment, we corrected the Y-axis into Shannon effective number of species in the revised version of the manuscript.

1. Jost, L. (2006). Entropy and diversity. *Oikos* 113, 363–375.
2. MACARTHUR, R.H. (1965). PATTERNS OF SPECIES DIVERSITY. *Biol. Rev.* 40, 510–533.

Figure 1B – include axis for MDS plot

In Figure 1B, we show MDS by performing a nonparametric transformation from the original multidimensional space into 2-space. The legend indicated the distance unit ($d=0.1$) – explaining the distances between points. The further away two points are the more dissimilar they are in multi-dimensional space, and conversely the closer two points are the more similar they are.

Lines 106 – authors have numerous comments about the function of genera that are significant in their analysis. However their analyses did not look at the functions of these bacteria. Suggest to remove these comments and move any description of potential functions and comparison to previously published literature to the discussion. Same for line 123 in comments about homocysteine.

Thank you for the feedback – we have moved these parts to the discussion as suggested.

Metabolomics

Line 126-138 – authors associate abundance of bacteria to bacterial metabolic profiles. While suggestive that there are causal links, there is not necessarily a linear relationship between genera abundance and bacterial function, please address and modify.

Thank you for the feedback, an excellent point! We have revised the respective part and added a sentence to highlight the potentially non-linear relationship between bacterial abundance and metabolite levels (=bacterial function). Highlighting this non-linear relationship, we have performed an additional analysis of gene copy numbers of the butyryl coenzyme A (CoA):acetate-CoA transferase (bcoA, involved in butyrate production) in rural and urban fecal samples using degenerate primers against bcoA. Here, higher levels of bcoA were detected in the urban Xhosa cohort despite similar fecal butyrate levels in both groups, exemplifying the non-linear relationship of bacterial abundance and function.

Virome

Include sample size for viral analyses – looks like there were only 8 urban participants included in the analysis? (fig 3D)

We apologize that this information was not noted in the earlier version of this manuscript. We have analyzed the viral communities in 28 samples, including nine urban samples. In addition, one sample was excluded as an outlier in the NMDS plot. We have now included this information in the manuscript.

The authors report on RNA viruses, but RNA viral analysis not described in supplementary material (line 145). If no RNA analysis was done, make this explicit in results and methodology.

We apologize for the confusion. The classifications are from Virosorter (<https://doi.org/10.1186/s40168-020-00990-y>). The reviewers are true that no RNA analyses were done. We now remove this part to avoid any confusion it may cause to the readers.

Describe detection of any eukaryotic viruses.

We did not find eukaryotic viruses. The bioinformatics tools that are developed for assigning taxonomy or predicting hosts for viral sequences are mainly made for the analysis of prokaryotic viruses. Using these tools is part of the standard procedure for virome analyses.

Authors used IPHoP to predict viral contig host assignment and BACPHLIP for replication cycles. Host assignment and description of replication is often poor. Can authors provide likelihood of accurate assignment to inform readers of the strengths of these associations/claims.

We agree with the reviewers that characterizing viral sequences from metagenomes is often challenging. We used iPHoP, which is an integrated machine-learning framework for maximizing host prediction of viral sequences. iPHoP can reliably predict host taxonomy at the genus level with a low false discovery rate (FDR <10%). BACPHLIP is also a robust tool commonly used to predict phage replication cycles in metagenomes. It has shown an accuracy of 98%. We are now adding this information to the material and methods section.

Lines 153-4 add p-values to comparisons.

The differences were not statistically significant and this was highlighted in the respective part.

Line 165 remove 'major'

This was removed as suggested.

Figure 3E, again, questionable validity of associating phage data with all dietary and metabolomic parameters. Particularly given lack of knowledge of temperate/lytic activity. Consider removing or moderating language describing associations to acknowledge possible shortcomings.

We understand the reviewer's concerns. We have added a few lines suggesting that further validations of the associations found in this study are necessary.

Authors use the term 'virulent' for several phages – how did they make this classification? Suggest to either define or remove

This is based on the predictions of the phages' replication cycles. Phages that can replicate only through the lytic cycle are referred to as virulent. We are now defining this in the manuscript.

Figure 3F – can the authors provide a better rationale for this comparison of phage host assignments between urban and rural subjects given the huge percentage of undefined viral contigs in combination with the expected accuracy of the phage-bacterial host assignments.

We understand the reviewer's concern. This is a limitation of most virome studies, mainly due to the under-representative public databases, and it was more evident in the samples obtained from the rural area. These data further highlight the differences in viral communities between the two regions and emphasize how little we know about them. We have now addressed this in the manuscript. We believe that the high level of unassigned phages would not significantly affect these comparisons, since all samples were subjected to this limitation.

Composition of food and skin microbiota

Line 177, modify text "is linked to urbanization of lifestyle" to be more reflective of strength of associations. e.g. there are differences between two cohorts residing in urban and rural settings.

Thank you for highlighting this point. We have changed this sub-title to "Composition of food and skin microbiota is different between rural and urban cohorts".

It is unclear how the microbiota of skin plays into the central research questions of this article. Why were the authors interested in the skin microbiota, and what hypotheses were they testing with their analysis?

We have included the microbiota analysis from hand skin swabs to generate preliminary insight into the effects of different environmental conditions on skin microbiota composition. As demonstrated by previous studies, different home environments along the urbanization gradient were associated with different abundance levels of skin bacteria (Mosites et al., 2017). Considering differences in hygiene conditions, kitchen settings and food processing techniques (that may involve contact of food to the skin of hands) between rural and urban households, this could provide an important environmental pool of bacteria or microbial stimuli with the potential to affect gut microbiota composition. We consider this an exploratory analysis given the low number of previously published data in the context of rural-urban transition in sub-Saharan populations. This was highlighted in the respective section of the results and discussion.

Mosites E, Sammons M, Otiang E, Eng A, Noecker C, Manor O, et al. Microbiome sharing between children, livestock and household surfaces in western Kenya. PLoS One. 2017;12(2):e0171017.

Figures – suggest to use one denotation for groups – either HE/HK or urban/rural

We have changed all denotations to rural/urban.

Discussions

Include a paragraph on the limitations of this study.

The authors show interesting associations between rural urban residence, dietary intake and microbiome. However their sample size is small, they do not look at risk of colorectal disease or NCD, and should remove claims of associations with these risks from the discussion. Additionally, “westernized” diet is a misleading term – differences in dietary intake varies enormously between individuals and is likely highly heterogenous across urban and rural settings, can the authors address inter-individual vs intergroup differences in their dietary analyses?

Thank you for the helpful feedback! We have added a new paragraph to the discussion that addresses the limitations of this study (see below). An additional response to these points is summarized in the next but one point (addressing the feedback starting with “Lines 209 - suggest to...”).

“Traditional food preparation is also relevant in the context of resistant starch, which is a dietary fiber rich in traditional high carbohydrate diets of rural South Africa, but not measured in this study as its content is variable and therefore difficult to measure with standard food composition tables³⁵. This also highlights the need to characterize dietary fibers in more detail in dietary analyses, accounting for fiber heterogeneity that may affect their functionalization by the gut microbiota^{51,4,52}. In this study, other factors (e.g., obesity, alcohol consumption, smoking, medication, different average age between cohorts) may impair the general extrapolation of observed biome signatures to urbanizing communities in low- or middle-income countries. Finally, the cross-sectional study layout, small sample size and focus on two study sites may not fully reflect the dynamic processes within urbanization, whereas the lack of colonic mucosal biopsies limits the conclusions with regard to CRC risk of study cohorts. Consistently, a recent systematic review highlighted that changes of the microbiota along the rural-urban gradient in Sub-Saharan Africa are heterogenous and multifactorial resulting in low universal patterns⁵³.”

Line 205: “despite the relatively high amounts of dietary fiber... their ‘Westernization of diet’ is likely being effective in promoting NCD risk, reflected by higher number of obesity cases”.... An alternative hypothesis is that there are differences in the microbiota because of gross baseline differences in the

cohort, such as obesity, energy expenditure, and environmental exposures that are broader than diet alone. Please address.

Thank you for the comment. Since we demonstrate associations of microbiome composition, obesity and dietary patterns, but no causal links, we cannot exclude the proposed hypothesis of obesity, energy expenditure or environmental exposures driving NCD risk in this setting. We have revised the sentence and highlighted the aspect of confounding factors in the paragraph on “limitations of the study” added to the discussion.

Line 209 – suggest to moderate this claim – authors don’t look at CRC risk. Suggest to remove onward suggestions that a ‘westernized diet’ in this cohort automatically gives a higher risk of CRC. Line 212 – this sentence suggests Fusobacteria in this cohort were linked to CRC incidence, which is not the case, suggest to revise.

Thank you for your comments. We do not claim that the diet of the urban cohort, which resembles a “Western” dietary pattern, would automatically result in a higher CRC risk. Similarly, we do not intend to claim that a higher relative abundance of Fusobacteria in feces of the urban study participants automatically results in higher CRC risk. Since we did not include the analysis of host mucosa samples (e.g., colonic biopsies) in this study, the functional and causal consequences of compositional differences between the urban and rural fecal microbiota remain to be elucidated. However, studies by us and others have demonstrated associations of “Westernized” diets, microbiome configuration and fecal metabolite patterns with CRC risk-associated host markers, so that assumptions regarding their effects on CRC risk can be discussed. In this study, these dietary and microbiome patterns differ between the two cohorts with the urban cohort being more similar to features of Western cohorts and NCD-associated patterns than their rural counterparts. Indeed, “Westernized diet” does not represent a clearly defined diet (similar to “dysbiosis” not reflecting a defined compositional state of the gut microbiota), so this term needs to be interpreted in the context of a dietary pattern ranging from “traditional/non-Western” to “modern/Western” without having clear definitions for the respective diet types. Within this relative range and in comparison with the rural cohort, the urban cohort is more likely to resemble a “modern/Westernized” diet based on the evidence provided in our study. In previous studies, we demonstrated that healthy rural African people show higher levels of short-chain fatty acids and lower levels of secondary bile acids in feces on their respective habitual diet compared to healthy African American or Alaska Native people, who have the world’s highest recorded CRC incidence and consume a low-fiber, high-fat diet (O’Keefe et al., Nature Communications 2015; Ocvirk et al., AJCN 2020). A diet switch among healthy volunteers from rural South Africa (switching to a high-fat low-fiber diet) and US African American participants (switching to a low-fat high-fiber diet) led to reciprocal changes in fecal short-chain fatty acid and bile acid levels as well as microbial genes involved in butyrate synthesis and bile acid transformation (O’Keefe et al., Nature Communications 2015). In this study, we also collected mucosal biopsies pre- and post-diet switch: critically, the diet switches correlated with consistent changes in mucosal markers associated with CRC risk, such as epithelial proliferation (Ki67) and immune cell infiltration (CD3 and CD68), suggesting a link between dietary patterns, microbiota configuration and diet-derived microbial metabolites.

Finally, there are studies indicating a low dietary diversity in South Africa with rural areas having lowest diet diversity scores (Shisana et al. South African National Health and Nutrition Examination Survey (SANHANES-1) HSRC Press; Cape Town, South Africa: 2013; Labadarios D. et al. How diverse is the diet

of adult South Africans? Nutr. J. 2011;10:33). Thus, even small cohorts like in this study may sufficiently cover the transition to an urban diet pattern that is characterized by higher dietary diversity. We have moderated the respective claims in the discussion section to highlight that patterns previously associated with NCD/CRC risk, but no clinically relevant host tissue/markers representing NCD/CRC risk were analyzed.

Methods

Dietary assessment – one time? How representative? Not Blinding?

The dietary assessment was performed at one time point and analyzed in non-blinded fashion. This information was added to the methods section and mentioned in the limitations of the study.

Line 309 – what is meant by ‘collections of food intake’?

Duplicates of all meals and drinks consumed over a 24-hour period were collected, homogenized, weighed and analyzed. We have revised the term to avoid potential misunderstandings.

Clarify number of samples collected per individual subject

One fecal sample was collected per individual subject at one time point and immediately distributed into four collection tubes that were frozen at -80 °C as soon as possible.

Urinary samples not described.

The description for urinary samples was added in the respective methods section.

Line 332 – title describes urine metabolic profiling, but not in description of activities

Thank you for the comments. We have revised the section accordingly.

Line 363 – RNA and DNA or only DNA? > Ali

We apologize for the confusion caused by the Virosorter results presented. No RNA analyses were conducted. We have addressed this issue in the manuscript.

How did authors deal with multiple sampling from one fecal sample? What were the numbers of freeze thaw cycles?

The fecal samples were immediately distributed to four collection tubes, one of them filled with DNA stabilization buffer (used for microbiome analysis). This way, multiple sampling from one tube/sample and freeze/thaw cycles were avoided.

Reviewer #2 (Remarks to the Author):

Introduction

While the importance of epidemiological transition in Africa – including lifestyle and diet – to health is understood, the topic remains under-studied. The microbiome is an important mediator and biomarker of the impact of change of diet, and so it is particularly useful to have studies which can describe how urbanisation impacts on the microbiome. Dysbiosis in the microbiome is linked to increased risk of colorectal cancer.

This paper studies compares the microbiome and metabolome of ≈ 40 amaXhosa, half urban, half rural.

Analysis of food intake is complex because the urban participants had a calorie intake 60% greater than the rural participants. So, urban participants had significantly more fibre than rural participants, but as a proportion of the diet, rural participants had more fibre (though this was not statistically significant). Urban dwellers has much more fat in total (highly statistically significant) and in proportion of diet (p -value, 0.06-0.08). Urban dwellers have a (non-statistically significant) greater total polyphenol consumption but a very statistically smaller normalised polyphenol consumption.

Analysis of the microbiome shows significantly smaller α -diversity in the urban dwellers and suggestive clustering when using β -diversity as a lens. Interesting and expected differences between key taxa can be found.

Study of the metabolome does not show distinct clustering though there are significant differences in key metabolites.

An analysis of the difference in the virome also reveals differences between the two groups. I thought particularly interesting that in the urban group, there were more viruses that are associated with bacteria often characterised as healthy.

Finally, there was an analysis of food and skin microbiome – interesting but only briefly reported.

Evaluation

1. The strengths of this research is that they have

- An urban and rural groups in the same country which are matched with respect to ancestry;
- Good characterisation of food intake (ffq for almost all, and detailed analysis of a sub-sample);
- Microbiome and metabolome analysis.

2. The interlinked analysis of the microbiome, metabolome and food is interesting

3. The authors don't identify limitations of the study. However, I think a weakness is the relatively small sample size and the insufficiently described groups. The authors say the sample size is in their experience big enough to detect differences between the groups. On the whole I buy this argument, but I think that a more serious problem is that in a complex society is whether in this respect the individuals who volunteered are representative of the population as a whole. See some detailed comment below.

Thank you for the feedback. We have added metadata (revised Table 1) to describe the two study cohorts in more detail and highlight potential confounding factors that may limit the data interpretation. We agree that the small sample size may not be representative of complex societies, which is a general issue of cross-sectional studies. This limitation was added to the discussion. However, the detailed biome and metabolome analyses may allow to derive general trends in the context of

nutrition transition from rural to urban areas in sub-Saharan Africa. Since there are studies indicating a low dietary diversity in South Africa with rural areas having lowest diet diversity scores (Shisana et al. South African National Health and Nutrition Examination Survey (SANHANES-1) HSRC Press; Cape Town, South Africa: 2013; Labadarios D. et al. How diverse is the diet of adult South Africans? Nutr. J. 2011;10:33), even small cohorts like in this study may sufficiently cover the transition to an urban diet pattern that is characterized by higher dietary diversity.

4. While at a high level the methodology seems sound, I would like to see more detail. For example, the bioinformatics analysis is not reproducible by anyone else – I could do a similar thing but now exactly what they did:

- For LDA the authors say “LDA ... was calculated as before ... [7]”. I assume it’s reference 7 of the supplementary since reference 7 of the main text doesn’t seem appropriate. Reference 7 of the supplementary is a general introduction to LEfSe (I don’t think authors in common so not sure what “as before” means).
- Ideally, the authors should describe exactly which scripts they used (and versions) with what parameters. If they have their own scripts they should be available Which versions of the database were used.

To respond to this concern, we revised the respective section and added additional details to the bacteriome and virome analysis methods section in the revised version of the manuscript as follows:

“Bacteriome Analysis

Raw data processing followed the IMNGS pipeline³ based on the UPARSE approach⁴. Sequences were demultiplexed, trimmed to the first base with a quality score < 3 and then paired. Reads with less than 300 (for skin swab samples: 250) and more than 600 nucleotides and paired reads with an expected error > 3 were excluded from the analysis. Additional trimming of remaining reads was done by removing 5 nucleotides on each end to avoid GC bias and non-random base composition. A table of zOTUs, respectively, was constructed by considering all reads before any quality filtering. ZOTUs (zero-radius OTUs) are valid operational taxonomic units that provide the maximum possible biological resolution⁵. A cut-off of 0.25% (e.g., deleting any zOTU not reaching this level in any sample) was used in order to avoid spurious zOTUs⁶. Taxonomy was assigned at an 80% confidence level using the SILVA ribosomal RNA gene database project⁷. Data analysis was conducted in the R programming environment using the Rhea R-package and NAMCO microbiome explorer⁸. To normalize the absolute read counts, the minimum sum counts were used within samples to calculate diversity. Sequencing depth was evaluated via rarefaction curves, and samples of low quality were excluded from the analysis (n=3 rural fecal samples). Group diversity was assessed using β -diversity based on generalized UniFrac distances or Bray-Curtis dissimilarity, and α -diversity was determined based on species richness and Shannon effective number of species diversity. P-values were computed using ANOVA on ranks and corrected for multiple comparisons following the Benjamini-Hochberg method. The impact of covariates on differences in the microbial profile across the entire study cohort was assessed through multivariate permutational analysis. This analysis was performed using the R function "adonis" from the vegan package version 2.5-6. The statistical significance was determined with a significance threshold of $P \leq 0.05$. For fecal samples, taxa with a prevalence of at least 30% in a given group were considered for statistical analysis. Differences between groups at genus level were examined using the linear discriminant analysis (LDA) effect size (LEfSe) method⁹, with the default settings available at <https://huttenhower.sph.harvard.edu/galaxy/root>. LEfSe employs a two-tailed nonparametric Kruskal-

Wallis test to assess the significance of variations between two groups. Differences in gut microbiota were considered significant, if they exhibited a p-value of < 0.05 and an LDA score (log₁₀) exceeding 3.

Correlation analysis between dietary components and fecal metabolites was performed using Pearson correlation method implemented in the Rhea pipeline. The centered log-ratio transformation is used to remove the compositional constraints from the taxonomic variables. In addition, taxonomic zeros (relative abundance of taxonomic variables with the value zero) were treated as missing data and were excluded from the calculation of correlations. Following this transformation of taxonomic variables, the table is centered and scaled, to adjust for differences in the offset and fold changes respectively, and the Pearson correlation for all pairs is calculated. The significance before and after FDR correction following the Benjamini-Hochberg method is reported together with the number of observations that support the correlation. The color of circles indicates the type of correlation (positive/negative) and radius of circles is proportional to the correlation.”

1. Jost, L. Entropy and diversity. *Oikos* **113**, 363–375 (2006).
2. MACARTHUR, R. H. PATTERNS OF SPECIES DIVERSITY. *Biological Reviews* **40**, 510–533 (1965).
3. Lagkourdos, I. et al. IMNGS: A comprehensive open resource of processed 16S rRNA microbial profiles for ecology and diversity studies. *Scientific Reports* **6**, 1–9 (2016).
4. Edgar, R. C. UPARSE : highly accurate OTU sequences from microbial amplicon reads. *Nature communications* **10**, (2013).
5. Edgar, R. C. UNOISE2: improved error-correction for Illumina 16S and ITS amplicon sequencing. *bioRxiv* 81257 (2016) doi:10.1101/081257.
6. Reitmeier, S. et al. Handling of spurious sequences affects the outcome of high-throughput 16S rRNA gene amplicon profiling. *ISME Communications* **1**, 31 (2021).
7. Yilmaz, P. et al. The SILVA and ‘all-species Living Tree Project (LTP)’ taxonomic frameworks. *Nucleic Acids Research* **42**, 643–648 (2014).
8. Lagkourdos, I., Fischer, S., Kumar, N. & Clavel, T. Rhea : a transparent and modular R pipeline for microbial profiling based on 16S rRNA gene amplicons. *PeerJ* (2017) doi:10.7717/peerj.2836.
9. Segata, N. et al. Metagenomic biomarker discovery and explanation. *Genome biology* **12**, R60–R60 (2011).

Viral metagenome Assembly and Analysis

Raw reads produced by sequencing were filtered using *fastp* to remove adaptors and low-quality bases. *Dedupe.sh* from the *bbmap* version 38.79 was used to remove duplicate reads. The remaining reads were then assembled using *metaSPAdes* with *k*-mer size set to 21,33,55,77,99. Contigs longer than 3000 bp from all samples were combined into a contig library (*cclib*). *CheckV* v0.8.1 was used to remove flanking host region of proviruses and assess the completeness and quality of generated contigs. To remove redundancy in the contig library, we clustered contigs into a non-redundant contig library (*nrclib*) using the “rapid genome clustering based on pairwise ANI” protocol in *CheckV*. The longest contig from each cluster sharing more than 95% identity and 80% coverage were selected as the representative in the *nrclib*. We then mapped the reads to *nrclib* using *minimap2*, and the abundance of each contig was calculated using *CoverM* v0.6.1 (<https://github.com/wwood/CoverM>). We used *VirSorter2* v2.2.3 to identify viral contigs (VCs) in *nrclib*. Contigs classified as categories 1 and 2 by *VirSorter2* were selected for further analyses. *DRAM-v* was used to annotate the identified VCs. The taxonomy assigned to these contigs using *vConTACT2* and a search against NCBI viral RefSeq database through the *MMSeqs2* taxonomy module. *IPHoP* was used to predict host of the VCs with default parameters, which is an integrated machine-learning framework for maximizing host prediction of viral sequences. *IPHoP* can reliably predict host taxonomy at the genus level with a low false discovery rate

(FDR <10%). VC replication cycle were predicted using BACPHLIP v0.9.6, which is a robust tool commonly used to predict phage replication cycles in metagenomes with shown accuracy of 98%. For the statistical analyses performed in R v.4.2.1 (R Core Team, 2022), the following packages were used: Deseq2 package for differential abundant analyses, the vegan package (<https://github.com/vegandevs/vegan>) was used to calculate the Shannon diversity (using the diversity() function), NMDS of Bray-Curtis dissimilarity (using metaMDS() function), and PERMANOVA for determining significant differences in Bray-Curtis dissimilarities (using adonis() function) using default parameters, corrplot package (<https://github.com/taiyun/corrplot>) for correlation analyses, Pearson correlation was used for analyzing the correlation between viral contigs, dietary components, and fecal metabolites, ggplot2 (<https://github.com/tidyverse/ggplot2>) package was used for creating graphics.

5. I did not see a commitment to making the data available (I searched as well). From what I understand there is no personally identifying data here and so there is not reason why the data should not be available in a public. The impact of this paper will be affected by whether the data is available. I understand this is not required for review, so I assume the authors will address this should the paper be accepted.

Thank you for raising this important point. Appropriate metadata and microbiome sequence data will be unified under a NCBI BioProject and sequence data will be submitted to the NCBI SRA (a respective paragraph was added), if the manuscript is accepted for publication.

6. In the discussion, much is correctly made about the transitional nature of the urban diet. To what extent is this transitional diet reflected in the microbiome – how does the microbiome differ with that found in other urbanised settings and what does this say about NCD risk and CRC risk in particular? There are some pointers on page 5 but more perhaps could be made.

We have revised the discussion and added references dealing with other cohorts affected by rural-urban transition. Interestingly, a recent systematic review highlighted that changes of the microbiota along the rural-urban gradient in Sub-Saharan Africa are heterogenous and depend on multiple factors (with diet being only one of them), resulting in low universal patterns (Paulo, L. S. et al. Urbanization gradient, diet, and gut microbiota in Sub-Saharan Africa: a systematic review. Frontiers in Microbiomes 2, (2023)) – this was very briefly discussed within the page limits.

7. Overall, I think the paper and the data produced make a valuable contribution to understanding epidemiological transition, and will be of interest to other researchers in the area.

Thank you for the feedback.

Other comments

8. The supplementary material should describe the cohorts in detail (you could fit one 1 page a table with urban participants on the left and rural on the right, showing age and BMI and sorted by sex and BMI).

The authors give standard error of the mean – I think that standard deviation would be more helpful to show range. I am doubtful given that you don't truly have random sampling that that shown standard error of mean is useful. To say that the average age of the rural cohort is 57 with a standard error of the mean being 2 when you have explicitly set limits on the age doesn't really tell us anything

about the population 2 sample. And given the population age profile a truly random sample would likely have a younger mean, I think.

Thank you for the comment. We have revised the table 1 showing additional metadata with regard to the study population and adding the standard error of mean as suggested. The average age between the two cohorts is significantly different (rural: 56.7 vs. urban: 50.4 years) and was addressed as potential (but unlikely) confounder of microbiome analyses.

The numbers don't seem to add up – in particular were participants excluded with > 40 BMI.

- Line 288 says a BMI ≤ 35 is an inclusion criterion.
- Line 294 says 24 rural and 21 urban participants were selected.
- Line 297 says 24 rural and 20 participants had dietary data assessed through questionnaire.
- Line 388 says 21 samples were used in both urban and rural – was this a decision to have an equal number or did some fail QC.
- Table 1 says that 1 of rural and 5 of the urban participants had a BMI > 35.
- Table 2 reports on all 44 samples for which dietary information is given.
- Table 1 lists 10 male and 10 female participants – did one participant not selfidentify as male or female?

Thank you for highlighting this error in table 1. The total number of participants from the urban area was not correct – there have been n=20 participants from the urban cohort. This was corrected in table 1 and throughout the manuscript. In some analyses (e.g., 16S rRNA gene sequencing, virome sequencing) some samples did not pass the QC and were excluded from the analysis – this was added to the respective analyses and figure legends.

9. line 76-78: I don't think it is true that NCDs are still relatively uncommon – even in some rural populations though obviously what counts as rural varies. Anyway, I think you need a citation to a recent (pref after 2021) high-quality paper that talks of the incidence of NCDs in South Africa here.

Thank you for the excellent comment. We performed literature research on that topic and agree with your assumption that the term “NCDs are relatively uncommon in South Africa” is not correct and does not factor in the rural-urban heterogeneity in NCD risk. A recent paper by Roomaney et al. (Int J Public Health, 2022) cited the second National Burden of Disease Survey (Pillay-van Wyk et al., Lancet Global Health, 2016) that reported mortality trends illustrating that in South Africa from 1997 to 2012, 43% of deaths were due to NCDs, 34% to HIV/AIDS and TB, 14% to other communicable diseases (and perinatal conditions, maternal causes and nutritional deficiencies) and 10% to injuries. Also, Vorster et al., (AJCN, 2014) reported sugars and sucrose-sweetened beverage consumption to be increased in both, urban and rural areas. Increased consumption was associated with increased NCD risk factors and the authors concluded that the nutrition transition has reached remote rural areas in South Africa. Finally, the Statistics South Africa Mortality Report (2017) demonstrated that NCDs accounted for 57.8% deaths in 2017. The respective sentence of the introduction was revised accordingly and three new references added to support this statement.

10. line 93: what does “distribution was within normal limits”. Do you mean that it is within the range normally found in western diets?

This statement was intended to point out that the distribution of macro-nutrients was within normal limits usually detected in Western diets. The statement was revised to avoid any potential misunderstanding.

11. line 99: the p-value for cholesterol in Table 2 is wrong.

Thank you for highlighting this error in table 2 – we have entered the correct value.

12. line 105/106 and throughout. I understand your use of HK and HE but I don't think they're the most obvious – rather use R and U for rural and urban

Thank you for this suggestion. We have revised the labeling according to your suggestions.

13. line 106: The claim that 1C makes is not obvious to me – caption needs to be longer

Thank you for the comment. We have revised the Figure Legend for Figure 1C and added more details for the explanation of the graph.

14. line 145-147: these differences look very different. Did you subject them to a formal statistical test?

Yes, however, the differences are not significant. This is mainly due to the interpersonal variation commonly seen in metagenomic data samples. We now mention these in the manuscript.

15. Line 157: I think better: “, which are...,” to “that are ...” (no commas)

Thank you for the comment – we have revised the section accordingly.

16. Line 258-260: This may be a point worth making before. Although you have good quality food data, the high level descriptors can be misleading. Not all fibre is the same, not all fat is the same

Thank you for this comment. We agree that the high level definition of dietary categories such as “fat” or “dietary fiber” can be misleading and affect evaluation and interpretation of results. This aspect was added to a new paragraph on limitations of the study in the discussion. To address this limitation of high level dietary analysis in future studies, we have collected additional food duplicate samples from the rural and urban cohorts and are currently establishing a collaboration with partners able to perform “foodomics”. Thus, in future studies we may be able to investigate food-derived chemical compounds in more detail that could help to better characterize the dietary intake.

17. Line 282: The word ‘homeland’ should be avoided in a South African context – too loaded a term. I am also concerned about the use of the word ethnicity since it is a complex term – I think better would be “self-identify as amaXhosa”.

Thank you for the comment – we have revised the section accordingly.

18. Line 283: centuries

This was revised as suggested.

19. Line 285: I think the population claim needs a good reference. Given the population of the whole province is about 7m, I find 2m for Khayalitsha as surprising. I see the number is highly contested in the Wikipedia article on Khayalitsha.

Thank you for the comment. We have contacted the Department of Statistics South Africa to ask for the updated numbers from the 2022 population census, but the data is not finalized, yet. Thus,

we have changed the number to the one given in the 2011 population census of about 392,000 and highlighted the Department of Statistics as reference.

20. Line 286-288: I think make it easier on the reader and put the inclusion criteria in the supplementary material. I had a look at reference 15. Did you rely on self-reporting for exclusion for diabetes?

We have moved the paragraph on inclusion and exclusion criteria to the supplementary material as suggested. Diabetes was excluded based on self-reporting – this information was added to the respective section.

21. Line 289 – A brief description of the district may be useful. In the context of SA's rapid transition there is considerable diversity in rural areas. To what extent do people in the district buy or produce food?

Thank you for the suggested additional information on the rural district. We have included a brief description of the district in the supplementary information as suggested.

22. Line 308: what kits were used for the gut microbiome collection?

We have used tubes filled with DNA-stabilizing buffer from Invitek Molecular Inc. The term "kit" may be misleading in this context and was changed to "collection tubes".

23. Lines 368-378: I think this discussion is at the right level of abstraction for the main body. But I think more detail is needed in the supplementary, otherwise this is not reproducible.

We have added additional details for the statistical analysis of bacteriome and virome data in the respective parts of the main manuscript and supplementary materials:

"Statistical analysis of bacteriome data:

P-values were computed using ANOVA on ranks and corrected for multiple comparisons following the Benjamini-Hochberg method. The impact of covariates on differences in the microbial profile across the entire study cohort was assessed through multivariate permutational analysis. This analysis was performed using the R function "adonis" from the vegan package version 2.5-6. The statistical significance was determined with a significance threshold of $P \leq 0.05$. For fecal samples, taxa with a prevalence of at least 30% in a given group were considered for statistical analysis. Differences between groups at genus level were examined using the linear discriminant analysis (LDA) effect size (LEfSe) method (with default settings available at <https://huttenhower.sph.harvard.edu/galaxy/root>). LEfSe employs a two-tailed nonparametric Kruskal-Wallis test to assess the significance of variations between two groups. Differences in gut microbiota were considered significant, if they exhibited a p-value of < 0.05 and an LDA score (\log_{10}) exceeding 3. Correlation analysis between dietary components and fecal metabolites was performed using Pearson correlation method implemented in the Rhea pipeline. The centered log-ratio transformation is used to remove the compositional constraints from the taxonomic variables. In addition, taxonomic zeros (relative abundance of taxonomic variables with the value zero) were treated as missing data and were excluded from the calculation of correlations. Following this transformation of taxonomic variables, the table is centered and scaled, to adjust for differences in the offset and fold changes respectively, and the Pearson correlation for all pairs is calculated. The significance before and after FDR correction following the Benjamini-Hochberg method is reported together with the number of observations that support the correlation. The color of circles indicates the type of correlation (positive/negative) and radius of circles is proportional to the correlation."

“Statistical analysis of virome data:

For the statistical analyses performed in R v.4.2.1 (R Core Team, 2022), the following packages were used: Deseq2 package for differential abundant analyses, the vegan package (<https://github.com/vegandevs/vegan>) was used to calculate the Shannon diversity (using the diversity() function), NMDS of Bray-Curtis dissimilarity (using metaMDS() function), and PERMANOVA for determining significant differences in Bray-Curtis dissimilarities (using adonis() function) using default parameters, corrplot package (<https://github.com/taiyun/corrplot>) for correlation analyses, Pearson correlation was used for analyzing the correlation between viral contigs, dietary components, and fecal metabolites, ggplot2 (<https://github.com/tidyverse/ggplot2>) package was used for creating graphics.”

24. In Table 1, I don't think using Xhosas as a plural is correct. Grammatically this should be adjectival describing “Sample Population” and so singular.

Thank you for the comment. We have revised the term to “rural/urban Xhosa cohort”.

Reviewer #3 (Remarks to the Author):

The study by Ramaboli MC et al. presents interesting findings that focus on linking the dietary intake and the fecal microbiome (bacteriome and virome)/metabolome profiles since multiple diseases including colorectal cancer (CRC) incidence associated with westernized high-fat diets by comparing healthy rural and urban Xhosa people in South Africa. It also supports the hypothesis that urbanization is associated with changes in lifestyle as part of economic development and is main drivers of non-communicable disease risk in sub-Saharan Africa. The manuscript is well-written, with clear descriptions and interpretations of the results.

To improve the quality of the manuscript, the following issues need to be addressed:

1. Line 110: Prevotella and Faecalibacterium were the main microbiotas highly abundant in rural individuals compared to urban individuals on a Westernized diet. A recent study showed a high abundance of Prevotella copri and Faecalibacterium prausnitzii in Kenyan healthy volunteers compared to CRC patients. This report supports the current findings (Obuya S, J Gastrointest Oncol. 2022;13(5):2282-2292. doi:10.21037/jgo-22-116).

Thank you highlighting the very good recent paper – we have included this reference in the third paragraph of the discussion.

2. Line 110& 218: In sufficient rationale/discussion related to findings that urban fecal samples also contained more Faecalibacterium phages that are commonly found in patients with IBD. This makes Faecalibacterium low abundant in urban compared to rural Xhosa fecal microbiota.

We thank the reviewer for the comments. We have now added a sentence related to this to the manuscript to discuss how a higher abundance of these phages could affect gut homeostasis by reducing their bacterial hosts.

3. Line 114: Bile acid secretion differs according to the nature of the consumed food that day. Was the fecal sample collected once and analyzed for the microbiome and the metabolite profiling?

Thank you for the feedback. The fecal samples were collected once and at a comparable time of the day for all study participants. The fecal samples were then distributed into four collection tubes, of which one contained DNA stabilization buffer – this aliquot was used for microbiome analysis, while the remaining samples were shipped frozen to the respective study site for further metabolomics analyses.

4. Line 120: It is believed that levels of bile acids (cholic acid, deoxycholic acid, and ursodeoxycholic acid) were higher in the feces of urban Xhosa people because they consume high-fat diets.

Thank you for the comment. Yes, the interpretation of data suggests that the levels of these bile acids are higher because of higher fat intake and the respective composition of the gut microbiota with more bacteria being adapted to bile acid exposure and biotransformation. We have added a sentence to the second paragraph of the discussion to point this out.

5. Line 210: If bacteriophages infect and kill bacteria and if Bacteroides were detected at high levels in diets rich in fat. In this case, bacteriodes-targeting viral contigs should be less in the urban Xhosa cohort eating high-fat diets. Please explain.

The interactions between phages and bacteria are highly complicated and do not always follow the proposed kill-the-winner dynamics, e.g., an increase in phage abundance would not always lead to a decrease in the abundance of the bacterial host. This has been explained by fluctuating-selection dynamics (<https://doi.org/10.1111/1574-6976.12072>), which could lead to long-term co-existence. This has been seen in Bacteroides phages (<https://doi.org/10.1186/s12915-021-01084-3>). We now have added a sentence related to this to the manuscript.

6. In methodology: Please provide the inclusion and exclusion criteria since the current study was done in CRC patients. It is also better to mention the alcohol consumption and the smoking status of individuals, if it is available since previous studies showed that they affect the microbiome.

Thank you for the comment. We have revised table 1 and added demographic data including data on smoking status, intake of medical drugs and alcohol consumption. These factors were also mentioned as potential confounders in a limitations paragraph that was added to the discussion. More details were added to the description on exclusion and inclusion criteria and this section was moved to the supplementary materials as suggested by one of the reviewers.

7. If it is possible, a confirmative qPCR for the main species found in the study would be an excellent validation.

We have performed two confirmative qPCR assays using specific primers for *Fusobacteria* and *B. wadsworthia*, which were both more abundant in urban Xhosa samples according to 16S rRNA gene sequencing. As shown in the graphs below, *Fusobacteria* and *B. wadsworthia* were detected at higher levels in urban compared with rural Xhosa fecal samples, confirming the sequencing results.

REVIEWERS' COMMENTS

Reviewer #1 (Remarks to the Author):

The authors have revised the article, "Diet Changes due to Urbanization in South Africa are Linked to Microbiome and Metabolome Signatures of Westernization and Increased Colorectal Cancer Risk"

(Note that, unfortunately, I am only able to access a merged file and not a track changes version of the article, limiting precise review)

The authors have carefully and extensively responded to reviewers' comments.

The following comments remain

The major remaining concern is the authors' persistent claim that their study findings are associated with increased colorectal cancer risk. This may be unintentional because the authors have addressed reviewer comments about this association in the results and discussion. However, the title and abstract continue to suggest that the study results directly associate with colorectal cancer risk within their cohort. The reviewer agrees that there is extensive literature that both diet and the microbiome associate with and are causal in the risk pathway and pathogenesis of colon cancer, however this article does not contain any colon cancer risk profiling and can therefore not make any claim of association. Strong suggestion to remove remaining leading statements in title, abstract (e.g. line 47) and results

Suggest that the authors submit a completed STROBE guideline checklist, incorporating the STROBE recommendations accordingly in to their manuscript for each study topic relevant cross-sectional cohort studies.

Title - suggest to remove or modify suggestions of causality in the title, replace with association and remove suggestion that diet/microbiome associate with CRC risk. Suggest to include the study design in the title, in line with the STROBE guidelines for cohort studies.

Language suggestions in Abstract

Line 44 abstract – change comparative to cross sectional

Line 46-48 suggest to remove or temper claims of bacterial function – the authors do not show metagenomic data supporting the metabolic function of bacteria identified.

Line 48 remove “tumor-promoting” this was not shown in the study

Line 53 clarify “reflected environmental conditions” which data do the authors refer to?

Line 82 Suggest to remove the term transitioning given the single timepoint used in the study

Table 1 – suggest to add percentage, suggest to move ‘antiretroviral therapy to under “HIV positive”

Table 2 – add samples sizes

Further, specifically add (in discussion) that antibiotic usage was not assessed as part of limitations

Discussion

It would be helpful if the authors can screen their discussion to better distinguish between what they have shown in their results and how they interpret their results.

Two examples include:

Line 207 – suggest to remove transition, transition not measured

Line 201-212 This conditioned a less diverse gut microbiota with bacterial compositions closer to Western cohorts and functionally adapted to Western diet stimuli (e.g., secondary bile acids). Suggest here to remove the word ‘conditioned’ (replace with, for example, associated) and suggest to add the word “likely” before closer (or provide data with comparison to Western cohort.)

Line 252 Suggest to add “may” show.....

Reviewer #2 (Remarks to the Author):

The authors' response to my comments has been more than adequate -- subject to this my previous review stands and the paper makes a significant contribution.

Reviewer #3 (Remarks to the Author):

The authors have addressed my concerns/comments and I do not have any further comments on their revised version.

Reviewer #1 (Remarks to the Author):

The authors have revised the article, "Diet Changes due to Urbanization in South Africa are Linked to Microbiome and Metabolome Signatures of Westernization and Increased Colorectal Cancer Risk"

(Note that, unfortunately, I am only able to access a merged file and not a track changes version of the article, limiting precise review)

The authors have carefully and extensively responded to reviewers' comments.

The following comments remain

The major remaining concern is the authors' persistent claim that their study findings are associated with increased colorectal cancer risk. This may be unintentional because the authors have addressed reviewer comments about this association in the results and discussion. However, the title and abstract continue to suggest that the study results directly associate with colorectal cancer risk within their cohort. The reviewer agrees that there is extensive literature that both diet and the microbiome associate with and are causal in the risk pathway and pathogenesis of colon cancer, however this article does not contain any colon cancer risk profiling and can therefore not make any claim of association. Strong suggestion to remove remaining leading statements in title, abstract (e.g. line 47) and results

Suggest that the authors submit a completed STROBE guideline checklist, incorporating the STROBE recommendations accordingly in to their manuscript for each study topic relevant cross-sectional cohort studies.

The STROBE guidelines checklist was added as suggested by the reviewer.

Title - suggest to remove or modify suggestions of causality in the title, replace with association and remove suggestion that diet/microbiome associate with CRC risk. Suggest to include the study design in the title, in line with the STROBE guidelines for cohort studies.

Thank you, we have revised the title and excluded "increased" in the statement related to "colorectal risk". Our study highlights changes in diet and microbiome-markers that have been previously associated with colorectal cancer risk. As described before, we do not intend to claim this study to be a CRC risk profiling for the investigated study cohorts and do not show causality given the observational character of the study – but when markers, which have been previously linked to CRC risk, are different between the two cohorts, this can be reflected in the title without being an overstated claim.

Language suggestions in Abstract

Line 44 abstract – change comparative to cross sectional

This was changed as suggested by the reviewer.

Line 46-48 suggest to remove or temper claims of bacterial function – the authors do not show metagenomic data supporting the metabolic function of bacteria identified.

This was changed as suggested.

Line 48 remove "tumor-promoting" this was not shown in the study

We agree with the reviewer that this was not shown in the study and we do not claim that this study investigates the tumor-promoting function of deoxycholic acid. But we and others have shown that this bile acid has tumor-promoting function, thus, we believe that this attribute can be used in this context. However, we have revised this statement to make clear that the CRC risk-associated function of this bile acid was analyzed in previous studies.

Line 53 clarify “reflected environmental conditions” which data do the authors refer to?
The statement was revised accordingly.

Line 82 Suggest to remove the term transitioning given the single timepoint used in the study
The term „ongoing transition“ does not relate to this particular study, but the general trend in South Africa. Since this study represents a snapshot within this „ongoing transition from traditional to urban lifestyle“, we believe that using this term highlights the relevance of such studies.

Table 1 – suggest to add percentage, suggest to move ‘antiretroviral therapy to under “HIV positive”
This was done as suggested.

Table 2 – add samples sizes
The sample sizes are given in the text under the table.

Further, specifically add (in discussion) that antibiotic usage was not assessed as part of limitations
Intake or treatment with antibiotics for the past 6 weeks was recorded and this information was already given as part of the exclusion criteria (supplementary information line 10).

Discussion

It would be helpful if the authors can screen their discussion to better distinguish between what they have shown in their results and how they interpret their results.

Two examples include:

Line 207 – suggest to remove transition, transition not measured

Here, „transition“ refers to „diet transition“ as a term that is not referring specifically to this study, but to the general trend. Since this study represents a snapshot within this general trend, this can (and should be) reflected by using the term „diet transition“ - also for putting the data of this study in a bigger picture, which should be an aim of a discussion paragraph.

Line 201-212 This conditioned a less diverse gut microbiota with bacterial compositions closer to Western cohorts and functionally adapted to Western diet stimuli (e.g., secondary bile acids).
Suggest here to remove the word ‘conditioned’ (replace with, for example, associated) and suggest to add the word “likely” before closer (or provide data with comparison to Western cohort.)
The sentence was revised accordingly.

Line 252 Suggest to add “may” show.....
The sentence was revised accordingly.